# MULTISCALE TRAINING OF CONVOLUTIONAL NEURAL NETWORKS

## ABSTRACT

Convolutional Neural Networks (CNNs) are the backbone of many deep learning methods, but optimizing them remains computationally expensive. To address this, we explore multiscale training frameworks and mathematically identify key challenges, particularly when dealing with noisy inputs. Our analysis reveals that in the presence of noise, the gradient of standard CNNs in multiscale training may fail to converge as the mesh-size approaches to 0, undermining the optimization process. This insight drives the development of Mesh-Free Convolutions (MFCs), which are independent of input scale and avoid the pitfalls of traditional convolution kernels. We demonstrate that MFCs, with their robust gradient behavior, ensure convergence even with noisy inputs, enabling more efficient neural network optimization in multiscale settings. To validate the generality and effectiveness of our multiscale training approach, we show that (i) MFCs can theoretically deliver substantial computational speedups without sacrificing performance in practice, and (ii) standard convolutions benefit from our multiscale training framework in practice.

## 1 INTRODUCTION

In this work, we consider the task of learning a functional $y(\mathbf{x}) = \phi(u(\mathbf{x}))$, where $\mathbf{x}$ is a position (in 2D $\mathbf{x} = (\mathbf{x}_1, \mathbf{x}_2)$ and in 3D $\mathbf{x} = (\mathbf{x}_1, \mathbf{x}_2, \mathbf{x}_3)$), $u(\mathbf{x}) \in \mathcal{U}$ and $y(\mathbf{x}) \in \mathcal{Y}$ are families of functions. To this end, we assume to have discrete samples from $\mathcal{U}$ and $\mathcal{Y}$, $(\mathbf{u}_i^h = u_i(\mathbf{x}_h), \mathbf{y}_i^h = \phi(u_i(\mathbf{x}_h)))$, $i = 1, \ldots, M$ associated with some resolution $h$. A common approach to learning the function is to parameterize the problem, typically by a deep network, and replace $\phi$ with a function $f(\cdot, \cdot)$ that accepts the vector $\mathbf{u}^h$ and learnable parameters $\boldsymbol{\theta}$ which leads to the problem of estimating $\boldsymbol{\theta}$ such that

$$\mathbf{y}_i^h \approx f(\mathbf{u}_i^h, \boldsymbol{\theta}), \quad i = 1, \ldots, M. \tag{1}$$

To evaluate $\boldsymbol{\theta}$, the following stochastic optimization problem is formed and solved:

$$\min_{\boldsymbol{\theta}} \ \mathbb{E}_{\mathbf{u}^h, \mathbf{y}^h} \ell \left( f(\mathbf{u}^h, \boldsymbol{\theta}), \mathbf{y}^h \right), \tag{2}$$

Where $\ell(\cdot, \cdot)$ is a loss function, typically mean square error. Standard approaches use variants of stochastic gradient descent (SGD) to estimate the loss and its gradient for different samples of $(\mathbf{u}^h, \mathbf{y}^h)$. In deep learning with convolutional neural networks, the parameter $\boldsymbol{\theta}$ (the convolutional weights) has identical dimensions, independent of the resolution. Although SGD is widely used, its computational cost can become prohibitively high as the mesh-size $h$ decreases, especially when evaluating the function $f$ on a fine mesh for many samples $\mathbf{u}_i^h$. This challenge is worsened if the initial guess $\boldsymbol{\theta}$ is far from optimal, requiring many costly iterations, for large data sizes $M$. One way to avoid large meshes is to use small crops of the data where large images are avoided, however, this can degrade performance, especially when a large receptive field is required for learning (Araujo et al., 2019)

**Background and Related Work.** Computational cost reduction can be achieved by leveraging different resolutions, a concept foundational to multigrid and multiscale methods. These methods have a long history of solving partial differential equations and optimization problems (Trottenberg et al., 2001; Briggs et al., 2000; Nash, 2000). Techniques like multigrid (Trottenberg et al., 2001) and algorithms such as MGopt (Nash, 2000; Borzi, 2005) and Multilevel Monte Carlo (Giles, 2015; 2008; Van Barel & Vandewalle, 2019) are widely used for optimization and differential equations.

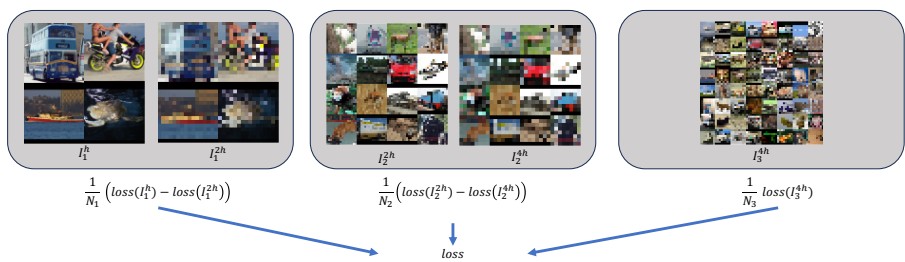

Figure 1: Illustration of our Multiscale-SGD, introduced in Section 2.

In deep learning, multiscale or pyramidal approaches have been used in image processing tasks such as object detection, segmentation, and recognition, where analyzing multiple resolutions is key (Scott & Mjolsness, 2019; Chada et al., 2022), reviewed in Elizar et al. (2022). Recent methods improve standard CNNs for multiscale computations by introducing specialized architectures and training methods. For instance, He & Xu (2019) uses multigrid methods in CNNs to boost efficiency and capture multiscale features, while Eliasof et al. (2023b) focuses on multiscale channel space learning, and van Betteray et al. (2023) unifies both. Li et al. (2020b) introduced the Fourier Neural Operator, enabling mesh-independent computations, and Wavelet-NNs were explored to capture multiscale features via wavelets (Fujieda et al., 2018; Finder et al., 2022; Eliasof et al., 2023a).

While often overlooked, it is important to note that these approaches, can be divided into two families of approaches that leverage multiscale concepts. The *first* is to learn parameters for each scale, and a separate set of parameters that mix scales, as in UNet (Ronneberger et al., 2015). The *second*, called *multiscale training*, enables the approximation of fine-scale parameters using coarse-scale samples (Haber et al., 2017; Wang et al., 2020; Ding et al., 2020; Ganapathy & Liew, 2008). The second approach aims to gain computational efficiency, as it approximates fine mesh parameters using coarse meshes, and it can be coupled with the first approach, and in particular with UNets.

**Our approach.** This work falls into the second category of multiscale training. We study multiscale algorithms that use coarse meshes to approximate high-resolution quantities, particularly the gradients of network parameters. Computing gradients on coarse grids is significantly cheaper than on fine grids, as noted in Shi & Cornish (2021). However, for efficient multiscale training, parameters on coarse and fine meshes must have "similar meaning," implying that both the loss and gradient on coarse meshes should approximate those on fine meshes. Specifically, the loss and gradient with respect to parameters should converge to a finite value as $h \to 0$. In this work, we show that standard CNN gradients may not converge as the mesh size $h$ approaches 0, suggesting CNNs under-utilize multiscale training. This motivates our development of mesh-free convolution kernels, whose values and gradients converge as $h \to 0$. Our approach builds on Differential-Operator theory (Wong, 2014) to create a family of learnable, mesh-independent convolutions for multiscale learning, resembling Fourier Neural Operators (FNO) (Li et al., 2020a) but with further expressiblity.

Our main contributions are: 1. Propose a new multiscale training algorithm, Multiscale SGD. 2. Analyze the limitations of standard CNNs within a multiscale framework. 3. Introduce a family of mesh-independent CNNs inspired by differential operators. 4. Validate our approach on benchmark tasks, showcasing enhanced efficiency and scalability.

## 2 MULTISCALE STOCHASTIC GRADIENT DESCENT

We now present the standard approach of training CNNs, identify its major computational bottleneck, and propose a novel solution called Multiscale Stochastic Gradient Descent (Multiscale-SGD).

**Standard Training of Neural Networks.** Suppose that we use a gradient descent-based method to train a CNN, with input resolution $h^1$ with trainable parameters $\boldsymbol{\theta}$. The $k$-th iteration reads:

$$\boldsymbol{\theta}_{k+1} = \boldsymbol{\theta}_k - \mu_k \mathbb{E}\left[\mathbf{g}(\mathbf{u}^h, \mathbf{y}^h, \boldsymbol{\theta}_k)\right],\qquad(3)$$

---

[1]In this paper, we define resolution $h$ as the pixel size on a 2D uniform meshgrid, where smaller $h$ indicates higher resolution. For simplicity, we assume the same $h$ across all dimensions, though different resolutions can be assigned per dimension.

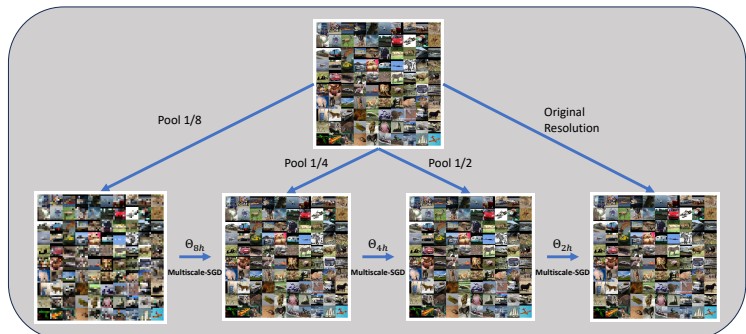

Figure 2: Illustration of the Full Multiscale Training described in Section 4.

where $\ell$ is some loss function (e.g., the mean-squared-error function), and the gradient of the loss with respect to the parameters is $\mathbf{g}(\mathbf{u}^h, \mathbf{y}^h, \boldsymbol{\theta}) = \boldsymbol{\nabla}\ell\left(f(\mathbf{u}^h, \boldsymbol{\theta}_k), \mathbf{y}^h\right)$. The expectation $\mathbb{E}$ is taken with respect to the input-label pairs $(\mathbf{u}^h, \mathbf{y}^h)$. Evaluating the expected value of the gradient can be highly expensive. To understand why, consider the estimation of the gradient obtained via the average over $\mathbf{g}$ with a batch of $N$ samples:

$$\mathbb{E}_{\mathbf{u}^h, \mathbf{y}^h}\left[\mathbf{g}(\mathbf{u}^h, \mathbf{y}^h, \boldsymbol{\theta})\right] \approx \frac{1}{N}\sum_i \mathbf{g}(\mathbf{u}_i^h, \mathbf{y}_i^h, \boldsymbol{\theta}_k). \tag{4}$$

This approximation results in an error in the gradient. Under some mild assumptions on the sampling of the gradient value (see Johansen et al. (2010)), the error can be bounded by:

$$\left\|\mathbb{E}\left[\mathbf{g}(\mathbf{u}^h, \mathbf{y}^h, \boldsymbol{\theta}_k)\right] - \frac{1}{N}\sum_i \mathbf{g}(\mathbf{u}_i^h, \mathbf{y}_i^h, \boldsymbol{\theta}^k)\right\|_2 \leq \frac{C}{\sqrt{N}}, \tag{5}$$

where $C$ is some constant. Clearly, obtaining an accurate evaluation of the gradient (that is, with low variance) requires sampling $\mathbf{g}$ across many data points with sufficiently high resolution $h$. This tradeoff between the sample size $N$ and the accuracy of the gradient estimation, is the costly part of training a deep network on high-resolution data. To alleviate the problem, it is common to use large batches, effectively enlarging the sample size. It is also possible to use variance reduction techniques (Anschel et al., 2017; Chen et al., 2017; Alain et al., 2015). Nonetheless, for high-resolution images, or 3D inputs, the large memory requirement limits the size of the batch. However, a small batch size can result in noisy, highly inaccurate gradients, and slow convergence (Shapiro et al., 2009).

## 2.1 Efficient Training with Multiscale Stochastic Gradient Descent

To reduce the cost of the computation of the gradient, we use a classical trick proposed in the context of Multilevel Monte Carlo methods (Giles, 2015). To this end, let $h = h_1 < h_2 < ... < h_L$ be a sequence of mesh step sizes, in which the functions $u$ and $y$ are discretized on. We can easily sample (or coarsen) $u$ and $y$ to some mesh $h_j, 1 \leq j \leq L$. We consider the following identity, based on the telescopic sum and the linearity of the expectation:

$$\mathbb{E}\left[\mathbf{g}^{h_1}(\boldsymbol{\theta})\right] = \mathbb{E}\left[\mathbf{g}^{h_L}(\boldsymbol{\theta})\right] + \mathbb{E}\left[\mathbf{g}^{h_{L-1}}(\boldsymbol{\theta}) - \mathbf{g}^{h_L}(\boldsymbol{\theta})\right] + \ldots + \mathbb{E}\left[\mathbf{g}^{h_1}(\boldsymbol{\theta}) - \mathbf{g}^{h_2}(\boldsymbol{\theta})\right], \tag{6}$$

where for shorthand we define the gradient of $\boldsymbol{\theta}$ with resolution $h_j$ by $\mathbf{g}^{h_j}(\boldsymbol{\theta}) = \mathbf{g}(\mathbf{u}^{h_j}, \mathbf{y}^{h_j}, \boldsymbol{\theta})$. The core idea of our Multiscale Stochastic Gradient Descent (Multiscale-SGD) approach, is that *the expected value of each term in the telescopic sum is approximated using a different batch of data with a different batch size*. This concept is demonstrated in Figure 1 and it can be summarizes as:

$$\mathbb{E}\left[\mathbf{g}^{h_1}(\boldsymbol{\theta})\right] \approx \frac{1}{N_L}\sum_i \mathbf{g}_i^{h_L}(\boldsymbol{\theta}) + \frac{1}{N_{L-1}}\sum_i \left(\mathbf{g}_i^{h_{L-1}}(\boldsymbol{\theta}) - \mathbf{g}_i^{h_L}(\boldsymbol{\theta})\right) + \ldots \tag{7}$$

$$+ \frac{1}{N_1}\sum_i \left(\mathbf{g}_i^{h_1}(\boldsymbol{\theta}) - \mathbf{g}_i^{h_2}(\boldsymbol{\theta})\right).$$

To understand why this concept is beneficial, we analyze the error obtained by sampling each term in Equation (7). Evaluating the first term in the sum requires evaluating the function $\mathbf{g}$ on the coarsest

mesh (i.e., lowest resolution) using downsampled inputs. Therefore, it can be efficiently computed, while utilizing a large batch size, $N_L$. Thus, following Equation (5), the approximation error of the first term in the Equation (7) can be bounded by:

$$\left\| \mathbb{E}\left[\mathbf{g}^{h_L}(\boldsymbol{\theta})\right] - \frac{1}{N_L}\sum_i \mathbf{g}_i^{h_L}(\boldsymbol{\theta}) \right\|^2 \leq \frac{C}{\sqrt{N_L}}. \tag{8}$$

Following this step, we need to evaluate the terms of the form

$$\mathbf{r}_j = \mathbb{E}\left[\mathbf{g}^{h_{j-1}}(\boldsymbol{\theta}) - \mathbf{g}^{h_j}(\boldsymbol{\theta})\right], \tag{9}$$

Similarly, this step can be computed by resampling, with a batch size $N_j$:

$$\hat{\mathbf{r}}_j = \frac{1}{N_j}\sum_i \left(\mathbf{g}_i^{h_{j-1}}(\boldsymbol{\theta}) - \mathbf{g}_i^{h_j}(\boldsymbol{\theta})\right), \tag{10}$$

for $j = 1, \ldots, L-1$. The key question is: what is the error in approximating $\mathbf{r}_j$ by the finite sample estimate $\hat{\mathbf{r}}_j$? Previously, we focused on error due to sample size. However, note that the exact term $\mathbf{r}_j$ is computed by evaluating $\mathbf{g}$ on two resolutions of the same samples and subtracting the results.

If the evaluation of $\mathbf{g}$ on different resolutions yields similar results, then $\mathbf{g}$ computed on mesh with step size $h_j$ can be utilized to approximate the gradient $\mathbf{g}$ on mesh a mesh with finer resolution $h_{j-1}$, making the approximation error $\hat{\mathbf{r}}_j$ small. Furthermore, assume that

$$\left\|\mathbf{g}_i^{h_{j-1}}(\boldsymbol{\theta}) - \mathbf{g}_i^{h_j}(\boldsymbol{\theta})\right\| \leq Bh_j^p \quad p > 0, \tag{11}$$

for some constant $B > 0$ and $p > 0$, both independent of the pixel-size $h_j$. Then, we can bound the error of approximating $\mathbf{r}_j$ by $\hat{\mathbf{r}}_j$, as follows:

$$\left\|\mathbf{r}_j - \hat{\mathbf{r}}_j\right\| \leq BC\frac{h_j^p}{\sqrt{N_j}}. \tag{12}$$

Note that, under the assumption that Equation (11) holds, the gradient approximation error between different resolutions decreases as the resolution increases (i.e., $h \to 0$). Indeed, sum of the gradient approximation error between subsequent resolutions (where each is defined in Equation (9)), where the approximation is obtained from the telescopic sum in Equation (7), can be bounded by:

$$e = C\left(\frac{1}{\sqrt{N_L}} + B\sum_{j=1}^{L-1}\frac{h_j^p}{\sqrt{N_j}}\right). \tag{13}$$

Let us look at an exemplary case, where $p = 1$ and $h_j = 2h_{j-1}$, i.e., the resolution on each dimension increases by a factor of 2 between input representations. In this case, the sampling error on the coarsest mesh contributes $N_L^{-1/2}$. It then follows that, it is also possible to have the same order of error by choosing $N_{L-1} = N_L/4$. That is, to obtain the same order of error at subsequent levels, only $1/4$ of the samples are required at the coarser grid compared to the finer one.

Following our Multiscale-SGD approach in Equation (7), the sample size needed on the finest mesh is reduced by a factor of $4^L$ from the original $N_L$ while maintaining the same error order, leading to significant computational savings. For instance, Table 3 shows that using 4 mesh resolution levels achieves an order of magnitude in computational savings compared to a single high-resolution gradient approximation.

Beyond these savings, Multiscale-SGD is easy to implement. It simply requires computing the loss at different input scales and batches, which can be done in parallel. Since gradients are linear, the loss gradient naturally yields Multiscale-SGD. The full algorithm is outlined in Algorithm 1.

## 2.2 MULTISCALE ANALYSIS OF CONVOLUTIONAL NEURAL NETWORKS

We now analyze under which conditions Multiscale-SGD can be applied for neural network optimization without compromising on its efficiency. Specifically, for Multiscale-SGD to be effective, the network output and its gradients with respect to the parameters at one resolution should approximate those at another resolution. Here, we explore how a network trained at one resolution, $h$, performs at

---

**Algorithm 1** Multiscale Stochastic Gradient Descent (Multiscale-SGD)

---

Set batch size to $N_L$ and sample, $N_L$ samples of $\mathbf{u}^{h_1}$ and $\mathbf{y}^{h_1}$
Pool $\mathbf{u}^{h_L} = \mathbf{R}^{h_1}_{h_L} \mathbf{u}^{h_1}, \quad \mathbf{y}^{h_L} = \mathbf{R}^{h_1}_{h_L} \mathbf{y}^{h_1}$
Set $loss = \ell(\mathbf{u}^{h_L}, \mathbf{y}^{h_L}, \boldsymbol{\theta})$
**for** $j = 1, ..., L$ (in parallel) **do**
    Set batch size to $N_j$ and sample, $N_j$ samples of $\mathbf{u}^{h_1}$ and $\mathbf{y}^{h_1}$
    Pool $\mathbf{u}^{h_j} = \mathbf{R}^{h_1}_{h_j} \mathbf{u}^{h_1}, \quad \mathbf{y}^{h_j} = \mathbf{R}^{h_1}_{h_j} \mathbf{y}^{h_1}$ and $\mathbf{u}^{h_{j-1}} = \mathbf{R}^{h_1}_{h_{j-1}} \mathbf{u}^{h_1}, \quad \mathbf{y}^{h_{j-1}} = \mathbf{R}^{h_1}_{h_{j-1}} \mathbf{y}^{h_1}$
    Compute the losses $\ell(\mathbf{u}^{h_j}, \mathbf{y}^{h_j}, \boldsymbol{\theta})$ and $\ell(\mathbf{u}^{h_{j-1}}, \mathbf{y}^{h_{j-1}}, \boldsymbol{\theta})$
    $loss \leftarrow loss - \ell(\mathbf{u}^{h_j}, \mathbf{y}^{h_j}, \boldsymbol{\theta}) + \ell(\mathbf{u}^{h_{j-1}}, \mathbf{y}^{h_{j-1}}, \boldsymbol{\theta})$
**end for**

---

| Resolution | 1/1024 | 1/512 | 1/256 | 1/128 | 1/64 | 1/32 |
|---|---|---|---|---|---|---|
| Noisy input | $9.3 \times 10^{-3}$ | $7.1 \times 10^{-3}$ | $4.9 \times 10^{-3}$ | $2.8 \times 10^{-3}$ | $2.0 \times 10^{-3}$ | $1.7 \times 10^{-3}$ |
| Smooth input | $1.2 \times 10^{-4}$ | $2.5 \times 10^{-4}$ | $5.1 \times 10^{-4}$ | $1.0 \times 10^{-3}$ | $1.9 \times 10^{-3}$ | $3.7 \times 10^{-3}$ |

Table 1: The discrepancy between two consecutive gradients for the problem in Equation (14). As the mesh resolution increases, the discrepancy reduces for *smooth* data, while it *increases* for noisy data.

a different resolution, $2h$. Specifically, let $f(\mathbf{u}^h, \boldsymbol{\theta})$ be a network that processes images at resolution $h$. The downsampled version, $\mathbf{u}^{2h} = \mathbf{R}^{2h}_h \mathbf{u}^h$, is generated via the interpolation matrix $\mathbf{R}^{2h}_h$. We aim to evaluate $f(\mathbf{u}^{2h}, \boldsymbol{\theta})$ on the coarser image $\mathbf{u}^{2h}$. A simple approach is to reuse the parameters $\boldsymbol{\theta}$ from the fine resolution, $h$. The following Lemma, Lemma 1, justifies such a usage under some conditions

**Lemma 1** (Convergence of standard convolution kernels.). *Let $\mathbf{u}^h, \mathbf{y}^h$ be continuously differentiable grid functions, and let $\mathbf{u}^{2h} = \mathbf{R}^{2h}_h \mathbf{u}^h, \quad \mathbf{y}^{2h} = \mathbf{R}^{2h}_h \mathbf{y}^h$ be their interpolation to a mesh with resolution $2h$. Let $\mathbf{g}^h$ and $\mathbf{g}^{2h}$ be the gradients of the function in Equation* (14) *with respect to $\boldsymbol{\theta}$. Then the difference between $\mathbf{g}^h$ and $\mathbf{g}^{2h}$ is*

$$\|\mathbf{g}^{2h} - \mathbf{g}^h\| = \mathcal{O}(h).$$

As can be shown in the proof (see Appendix B), the convergence of the gradient depends on the amount of noise in the data. However, as we now show in Example 1, for noisy problems, this method may not always yield the desired outcome.

**Example 1.** *Assume that $f$ is a 1D convolution, that is $f(\mathbf{u}^h, \boldsymbol{\theta}) = \mathbf{u}^h \star \boldsymbol{\theta}$, and a linear model*

$$\ell_h(\boldsymbol{\theta}) = \frac{1}{n}(\mathbf{u}^h \star \boldsymbol{\theta})^\top \mathbf{y}^h, \tag{14}$$

*where $\mathbf{y}^h$ is the discretization of some function $y$ on the fine mesh, $h$. We use a 1D convolution with kernel size 3, whose trainable weight vector is $\boldsymbol{\theta} \in \mathbb{R}^{3 \times 1}$ and compute the gradient of the function $y$ on different meshes. The loss function on the $i$-th mesh can be written as*

$$\ell_{2^i h}(\boldsymbol{\theta}) = \frac{1}{n_i}(\mathbf{R}^{2^i h}_h \mathbf{u}^h \star \boldsymbol{\theta})^\top (\mathbf{R}^{2^i h}_h \mathbf{y}^h), \tag{15}$$

*where $\mathbf{R}^{2^i h}_h$ is a linear interpolation operator that takes the signal from mesh size $h$ to mesh size $2^i h$. We compute the difference between the gradient of the function on each level and its norm*

$$\delta g = \|\boldsymbol{\nabla}_{\boldsymbol{\theta}} \ell_{2^i h} - \boldsymbol{\nabla}_{\boldsymbol{\theta}} \ell_{2^{i+1} h}\|^2. \tag{16}$$

*We conduct two experiments. In the first, we use smooth inputs, and in the second, we add Gaussian noise $N(0, 1)$ to the inputs. We compute the values of $\delta g$, and report it in Table 1. The experiment demonstrates that for noisy inputs, the gradient with respect to the convolution parameters grows in tandem with the mesh. Thus, using convolution in its standard form will not be useful in reducing the error using the telescopic sum in Equation* (6) *as described above.*

The analysis suggests that for smooth signals, standard CNNs can be integrated into Multilevel Monte-Carlo methods, which form the basis of our Multiscale-SGD. However, in many cases, inputs may be noisy or contain high frequencies, leading to large or unbounded gradients across mesh resolutions. This explains the lack of convergence of standard CNNs in Example 1. High-frequency inputs can hinder learning in multiscale frameworks like Multiscale-SGD. To address this, we propose a new convolution type in Section 3 that theoretically alleviates this issue.

# 3 MULTISCALE TRAINING WITH MESH-FREE CONVOLUTIONS

In Section 2.2, we analyzed standard convolutional kernels when coupled with our Multiscale-SGD training approach. As we have shown, standard convolutional kernels can scale poorly in the presence of noisy data. To address this limitation, we now propose an alternative to standard convolutions through the notion of 'mesh-free' or weakly mesh-dependent convolution kernels.

## 3.1 MESH-FREE CONVOLUTIONS

We introduce Mesh-Free Convolutions (MFCs), which define convolutions in the functional space independently of any mesh, then discretize them on a given mesh. Unlike standard convolution kernels tied to specific meshes, MFCs converge to a finite value as the mesh refines, overcoming the limitations of standard kernels discussed in Section 2.2. One way to achieve this goal is to define the convolution in functional space directly by an integral, that is

$$v(\mathbf{x}) = \int_\Omega \mathcal{K}_{\boldsymbol{\xi}}(\mathbf{x} - \mathbf{x}')u(\mathbf{x}')d\mathbf{x}', \tag{17}$$

where the kernel function $\mathcal{K}_{\boldsymbol{\xi}}$ is, parametrized by learnable parameters $\boldsymbol{\xi}$. This general convolution can be clearly defined on any mesh and is basically mesh-free. The question remains, how to parametrize the kernel. Clearly, there are many options for this choice. A too restrictive approach leads to limited expressivity of the convolution. Our goal and intuition is to generate kernels that are commonly obtained by trained CNNs. Such filters are rarely random but contain some structure (Goodfellow et al., 2016). To this end, we use the interplay between the solution of partial differential equations (PDEs) and convolutions (Evans, 1998). In particular, the solution of parabolic PDEs (that is, the heat equation) is a kernel that exhibits the behavior that is often required from CNN filters.
Let $u$ be input functions (that is, $u$ can have more than a single channel, or so-called vector function) and let $v = \mathcal{C}(\boldsymbol{\xi})u$ be the application of the mesh-free convolution is parameterized by $\boldsymbol{\xi}$. We obtain a mesh-independent convolution by composing two processes. The first is a trainable matrix $\mathbf{R} \in \mathbb{R}^{d \times c}$ which is a $1 \times 1$ convolution, that embeds $u$ into a subspace $\tilde{v}$, yielding the initial condition of the PDE defined in Equation (18). The second is the solution of the parabolic PDE of the form below, which maps the initial condition $\tilde{v}(0, \mathbf{x})$ into $\tilde{v}(\tau, \mathbf{x})$

$$\tilde{v}_t = \mathcal{L}(\boldsymbol{\alpha})\tilde{v}, \quad \tilde{v}(t=0) = \mathbf{R}u, \quad t \in [0, \tau], \qquad v = (\gamma_0 + [\gamma_x, \gamma_y]^\top \boldsymbol{\nabla})\tilde{v}(\mathbf{x}, \tau). \tag{18}$$

Finally, the output $v(\mathbf{x})$ is obtained by taking a directional derivative of $\tilde{v}(\tau, \mathbf{x})$ in a direction $\boldsymbol{\gamma}$. All the processes above are encompassed by the mesh-free convolution operator $\mathcal{C}(\boldsymbol{\xi})$. The operator $\mathcal{L}(\boldsymbol{\alpha}) = \text{diag}\left(\mathcal{L}(\boldsymbol{\alpha}_1), \ldots, \mathcal{L}(\boldsymbol{\alpha}_d)\right)$ is an Elliptic Differential Operator, that we define as

$$\mathcal{L} = \alpha_{xx}\frac{\partial^2 v}{\partial x^2} + 2\alpha_{xy}\frac{\partial^2 v}{\partial xy} + \alpha_{yy}\frac{\partial^2 v}{\partial y^2}. \tag{19}$$

The trainable parameters are $\boldsymbol{\xi} = [\tau, \boldsymbol{\alpha}, \boldsymbol{\gamma}] \in \mathbb{R}^7$, where $\tau$ is a scalar, $\boldsymbol{\gamma} = [\gamma_0, \gamma_x, \gamma_y]$, and $\boldsymbol{\alpha} = [\alpha_{xx}, \alpha_{xy}, \alpha_{yy}]$. We enforce $\alpha_{xx}, \alpha_{yy} > 0$ and $\alpha_{xx}\alpha_{yy} - \alpha_{xy}^2 > 0$, ensuring $\mathcal{L}$ is elliptic and invertible (Trottenberg et al., 2001). Integrating the PDE over time $[0, \tau]$ and differentiating in the direction $[\gamma_x, \gamma_y]$ effectively applies the convolution to the input $u$. The parameter $\tau$ acts as a scaling factor: increasing it widens and smooths the kernel, while as $\tau \to 0$, the kernel becomes more localized. To visualize our convolution, we sample random parameters $\boldsymbol{\xi}$ and plot the resulting MFC filters in Figure 3. By varying $\boldsymbol{\xi}$, we generate a range of convolutional kernels, from local to global, with different orientations. Here, $\alpha$ controls orientation, $\gamma$ sets the lobe direction, and $\tau$ determines the kernel width. The figure demonstrates that our choice of convolutions yields trainable filters to the ones that are typically obtained by CNN (Goodfellow et al., 2016). These filters can change their field of view by changing $t$, change their orientation by changing $\boldsymbol{\xi}$, and change their side lobes' strength and direction by changing $\boldsymbol{\gamma}$.

**Computing MFCs on a Discretized Mesh.** Our MFCs are defined by the solution of Equation (18) and are continuous functions. Here, we discuss their computation on a discretized mesh for application to images. For a rectangular domain $\Omega$, we use periodic boundary conditions and the Fast Fourier Transform (FFT) to compute the convolution. Let $\mathbf{u}^h$ be a discrete tensor. First, we compute its Fourier transform, $\hat{\mathbf{u}}^h = \mathbf{F}\mathbf{u}^h$, and use a frequency mesh to obtain the 2D frequency grid $\mathbf{k}_x$ and $\mathbf{k}_y$. The discrete symbol $\mathcal{L}(\mathbf{k}_x, \mathbf{k}_y)$ is a 3D tensor operating on each channel, similar to standard

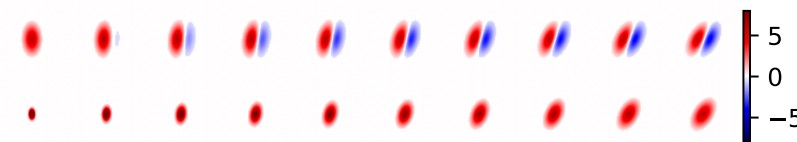

Figure 3: **An illustration of Mesh-Free Convolutions (MFCs) is provided to showcase the effect of parameter variations. First row:** The parameters are set as $\alpha_{xx} = 1$, $\alpha_{yy} = 3$, while $\alpha_{xy}$ varies from 0 to 1, and $\gamma_x$ changes from 0 to 1. **Second row:** The parameters are set as $\gamma_x = \gamma_y = 0$, $\gamma_0 = 1$, with $\alpha_{xy}$ varying from 0 to 1, and $t$ changing from 0 to 1. These visualizations highlight the flexibility and expressiveness of MFCs in adapting to different parameter configurations.

2D filters. We then compute a point-wise product and use the inverse FFT to return to image space. Interestingly, given some fixed mesh of size $h$ and a regular 2D convolution $\mathbf{C}$ it is always possible to build an MFC, $\mathbf{C}_{\boldsymbol{\xi}}$, such that they are equivalent when evaluated **on that mesh**. The proof is given in Appendix B, and it implies that MFCs have at least the same expressiveness as standard convolutions.

**Computational Complexity.** MFC first applies a standard pointwise operation to expand a $d$-channel tensor into a $c$-channel tensor. The main cost comes from the spatial operations on the $c$ channels, particularly the FFT, which has a cost of $n \log(n)$, where $n$ is the number of pixels. With $c$ channels, the total cost becomes $\mathcal{O}(cn \log(n))$. As other operations in the network scale linearly with $n$, the overall cost is dominated by the FFT.

### 3.2 FOURIER ANALYSIS OF MESH-FREE CONVOLUTIONS

We now analyze the ability of MFCs to produce consistent gradients across scales, enabling effective use of Multiscale-SGD from Section 2. Unlike standard convolutions, MFCs are defined as a continuous function, not tied to a specific mesh. This allows us to compute gradients directly with respect to the function, equivalent to computing gradients at the finest resolution, as $h \to 0$.
Following the analysis path shown in Section 2.2, consider the linear loss obtained by these MFCs when discretizing the functions $u$ and $y$. In the case of functions, such as with our MFCs, the inner product in Equation (14) is replaced by an integral that reads:

$$\ell^*(\boldsymbol{\xi}) = \int_0^1 y(x)\mathcal{C}(\boldsymbol{\xi})u(x)\,dx. \tag{20}$$

We study the behavior of our MFC, denoted by $\mathcal{C}(\boldsymbol{\xi})$, on a function $u(x)$, by computing its symbol. Because the heat equation is linear, the symbol of $\mathcal{C}(\boldsymbol{\xi})$ is given by (see Trottenberg et al. (2001)):

$$\mathcal{L}(k_x, k_y) = (\gamma_0 + ik_x\gamma_x + ik_y\gamma_y) \exp\left(-\tau(\alpha_{xx}k_x^2 + 2\alpha_{xy}k_xk_y + \alpha_{yy}k_y^2)\right). \tag{21}$$

We now provide a Lemma that characterizes the behavior of our MFCs and, in particular, shows that unlike with traditional convolutional operations, our MFCs yield consistent gradient approximations independently of the mesh resolution – thereby satisfying the assumptions of our Multiscale-SGD training approach. The proof is provided in Appendix B.

**Lemma 2** (MFCs yield consistent gradients independently of mesh resolution.)**.** *Let the loss be defined as Equation* (20) *and let $u(x)$ and $y(x)$ be any two integrable functions. Then, $-\infty < \ell^* < \infty$, that is, the loss of any two integrable functions is a finite number. Furthermore, let $\nabla_{\boldsymbol{\xi}}\ell^*$ be the gradient of the loss with respect to its parameters. Then we have that $-\infty < \nabla_{\boldsymbol{\xi}}\ell^* < \infty$, that is, its gradient is a vector with finite values.*

The property described in Lemma 2 carries forward to any discrete space, i.e., chosen mesh. Namely, by discretizing our MFCs on a uniform mesh with mesh-size $h$ we obtain its discrete analog:

$$\ell(\boldsymbol{\xi}) = \frac{1}{n}(\mathbf{y}^h)^\top (\mathcal{C}(\boldsymbol{\xi})\mathbf{u}^h). \tag{22}$$

Furthermore, using the discrete Fourier Transform $\mathbf{F}$, we obtain that

$$\ell^h(\boldsymbol{\xi}) = \frac{1}{n}(\mathbf{F}\mathbf{y}^h)^\top \left(\mathcal{L}(k_x, k_y)\mathbf{F}\mathbf{u}^h\right) = \frac{1}{n}\sum_{k_x,k_y} \mathcal{L}(k_x, k_y) \odot \widehat{\mathbf{y}}^h(k_x, k_y) \odot \widehat{\mathbf{u}}^h(k_x, k_y). \tag{23}$$

| Resolution | 1/1024 | 1/512 | 1/256 | 1/128 | 1/64 | 1/32 |
|---|---|---|---|---|---|---|
| Noisy input | $1.6 \times 10^{-4}$ | $3.2 \times 10^{-4}$ | $6.1 \times 10^{-4}$ | $1.3 \times 10^{-3}$ | $2.9 \times 10^{-3}$ | $5.9 \times 10^{-3}$ |
| Smooth input | $1.1 \times 10^{-4}$ | $2.1 \times 10^{-4}$ | $4.2 \times 10^{-4}$ | $7.8 \times 10^{-4}$ | $1.5 \times 10^{-3}$ | $3.1 \times 10^{-3}$ |

Table 2: The discrepancy between two consecutive gradients for the example problem Equation (14) using mesh-free convolutions. The discrepancy is reduced for both noisy and smooth data.

---

**Algorithm 2** Full-Multiscale-SGD

---

Randomly initialize the trainable parameters $\boldsymbol{\theta}_*^H$.
**for** $j = 1, ..., L$ **do**
  Set mesh size to $h_j = 2^{L-j}h$ and $\boldsymbol{\theta}_0^{h_j} = \boldsymbol{\theta}_*^H$.
  Solve the optimization problem on mesh $h_j$ using SGD and Algorithm 1 for $\boldsymbol{\theta}_*^{h_j}$.
  Set $\boldsymbol{\theta}_*^H \leftarrow \boldsymbol{\theta}_*^{h_j}$.
**end for**

---

As both $\widehat{\mathbf{y}}^h$ and $\widehat{\mathbf{u}}^h$ are discrete Fourier transforms of the functions $y$ and $u$, they converge to their continuous counterparts as $h \to 0$. Thus, $\ell^h$ converges to $\ell^*$ as $h \to 0$, leading to a key result: *unlike standard convolutions, the gradients of our MFCs converge to a finite value as the mesh refines, making them suitable for multiscale techniques like Multiscale-SGD, summarized in Lemma 3.*

**Lemma 3** (MFCs gradients converge as the mesh resolution refines.). *Let $\ell^h(\boldsymbol{\xi})$ be define by Equation* (22). *Then at the limit,*

$$\lim_{h \to 0} \ell = \ell^* \quad \text{and} \quad \lim_{h \to 0} \boldsymbol{\nabla}_{\boldsymbol{\xi}} \ell = \boldsymbol{\nabla}_{\boldsymbol{\xi}} \ell^*.$$

To demonstrate this property of the mesh-free convolution, we repeat Example 1, but this time with the mesh-free convolutions. The results are reported in Table 2.

## 4  THE FULL-MULTISCALE ALGORITHM

To leverage a multiscale framework, a common method is to first solve on a coarse mesh and interpolate it to a finer mesh, a process called mesh homotopy (Haber et al., 2007). When training a neural network, optimizing the weights $\boldsymbol{\theta}$ on coarse meshes requires fewer iterations to refine on fine meshes (Borzì & Schulz, 2012). Using Multiscale-SGD (Algorithm 1), the number of function evaluations on fine meshes is minimized. This resolution-dependent approach is summarized in Full-Multiscale-SGD (Borzì & Schulz, 2012) (Algorithm 2) and illustrated in Figure 2.

**Convergence rate of Full-Multiscale-SGD.** To further understand the effect of the full multiscale approach, we recall the stochastic gradient descent converge rate. For the case where the learning rate converges to 0, we have that after $k$ iterations of SGD, we can bound the error of the loss by

$$\left| \mathbb{E}\left[\ell(\boldsymbol{\theta}_k)\right] - \mathbb{E}\left[\ell(\boldsymbol{\theta}_*^h)\right] \right| \le C_0 \frac{C}{k}, \tag{24}$$

where $C$ is a constant, $\boldsymbol{\theta}_*^h$ is the parameter that optimizes the expectation of the loss, and $C_0$ is a constant that depends on the initial error at $\boldsymbol{\theta}_0$. Let $\boldsymbol{\theta}_*^H$ and $\boldsymbol{\theta}_*^h$ be the parameters that minimize the expectation of the loss for meshes with resolution $H > h$ and assume that $\|\boldsymbol{\theta}_*^H - \boldsymbol{\theta}_*^h\| \le \gamma H$, where $\gamma$ is a constant. This assumption is justified if the loss converges to a finite value as $h \to 0$. During the Full-Multiscale-SGD iterations (Algorithm 2), we solve the problem on the coarse mesh $H$ to initialize the fine mesh $h$. Thus, after $k$ steps of SGD on the fine mesh, we can bound the error by:

$$\left| \left[\mathbb{E}\ell(\boldsymbol{\theta}_k)\right] - \mathbb{E}\left[\ell(\boldsymbol{\theta}_*^h)\right] \right| \le \gamma H \frac{C}{k}. \tag{25}$$

Requiring that the error is smaller than some $\epsilon$, renders a bounded number of required iterations $k \approx \gamma C \frac{H}{\epsilon}$. Since in Algorithm 2, $H = 2^j h$, the number of iterations for a fixed error $\epsilon$ is halved at each level, the iterations on the finest mesh are a fraction of those on the coarsest mesh. As shown in Table 3, multiscale speeds up training by a factor of $\sim 160$ compared to standard single-scale training.

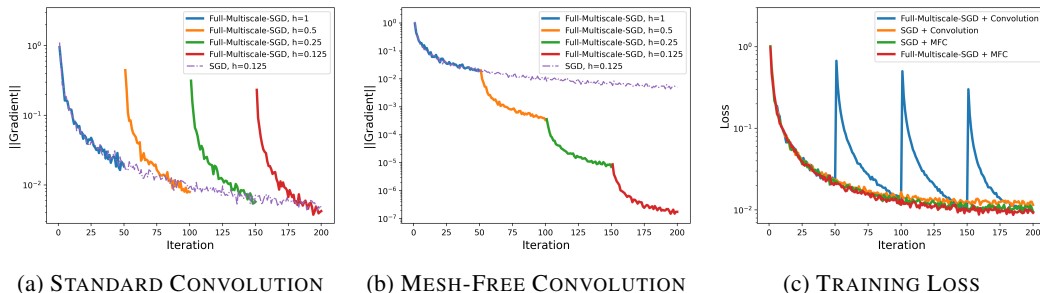

(a) STANDARD CONVOLUTION      (b) MESH-FREE CONVOLUTION      (c) TRAINING LOSS

Figure 4: Training on a Single Scale vs. Full-Multiscale-SGD (Ours). (a) uses standard convolutions, and (b) shows Mesh-free convolutions (MFCs). Standard convolutions struggle with varying input resolutions, as seen in the gradient increase, while MFCs handle continuous training with varying resolutions. (c) plots training loss for standard and multiscale strategies with both convolution types, showing that standard convolutions are unsuitable for multiscale training, whereas MFCs enable it.

## 5  EXPERIMENTAL RESULTS

In this section, we evaluate the empirical performance of our training strategies: *Multiscale-SGD* (Algorithm 1) and *Full-Multiscale-SGD* (Algorithm 2), compared to the standard CNN training with *SGD*. We demonstrate these strategies on architectures like ResNet (He et al., 2016) and UNet (Ronneberger et al., 2015), and their MFC versions, where we replace standard convolutions with our mesh-free convolutions (Section 3.1). We denote these modified architectures as *MFC-X*, where X is the original network architecture (e.g., MFC-ResNet and MFC-UNet). Additional details on experimental settings, hyperparameters, architectures, and datasets are provided in Appendix E.

**Research Questions.** We seek to address the following questions: 1. How effective are standard convolutions with multiscale training? 2. Do mesh-free convolutions (MFCs) perform as predicted theoretically in a multiscale setting? 3. Can multiscale training be broadly applied to typical CNN tasks, and how much computational savings does it offer compared to standard training?

**Metrics.** To address our questions, we focus on performance metrics (e.g., MSE or SSIM) and the computational effort for each method. As a baseline, we use a standard ResNet or UNet trained on a single input resolution. We define a *work unit* (#WU ) as one application of the model on a single image at the finest mesh (i.e., original resolution). In multiscale training, this unit decreases by a factor of 4 with each downsampling. We then compare #WU  across training procedures.

**Blur Estimation.** We begin by solving a quadratic problem to validate the theoretical concepts discussed in Sections 2.2 and 3.2. Specifically, we estimate a blurring kernel directly from data (see Nagy & Hansen (2006) for the definition). We assume both the image $\mathbf{u}$ and its blurred version $\mathbf{y}$ are available on a $256^2$ fine mesh. The relationship is linear, given by $\mathbf{y} = \mathbf{G} \star \mathbf{u} + \epsilon$, where $\mathbf{G}$ is the blurring kernel. Given $\mathbf{u}$ and $\mathbf{y}$, we solve the optimization problem $\min_{\mathbf{G}} \frac{1}{2}\mathbb{E}_{\mathbf{u},\mathbf{y}}\|\mathbf{y} - \mathbf{G} \star \mathbf{u}\|^2$.

This is a quadratic stochastic optimization problem for the kernel $\mathbf{G}$. We simulate $\mathbf{u}$ as a Random Markov Field (Tenorio et al., 2011) using a random $7 \times 7$ blurring kernel for $\mathbf{G}$. The problem is solved with standard CNNs and MFCs, comparing SGD, Multiscale-SGD, and Full-Multiscale-SGD strategies. As seen in Figure 4, for standard convolutions, the loss increases with mesh refinement, making multiscale approaches ineffective. However, our MFCs maintain stable loss across meshes, making the multiscale process efficient, with the final 50 iterations on the finest mesh. We measure computational work by running 200 iterations on the fine mesh with SGD, Multiscale-SGD, and Full-Multiscale-SGD, and early-stopping when the test loss plateaus. Table 3 shows that using Multiscale-SGD yields significant savings, and Full-Multiscale-SGD offers even further savings.

**Image Denoising.** Here, one assumes data of the form $\mathbf{u}^h = t\mathbf{y}^h + (1 - t)\mathbf{z}$ where $\mathbf{y}^h$ is some image on a fine mesh $h$ and $\mathbf{z} \sim N(0, \mathbf{I})$ is the noise. The noise level $t \in [0, 1]$ is chosen randomly. The goal is to recover $\mathbf{u}^h$ from $\mathbf{y}^h$. The loss to be minimized is $loss(\boldsymbol{\theta}) = \frac{1}{2}\mathbb{E}_{\mathbf{y}^h,\mathbf{u}^h,t}\|f(\mathbf{u}^h, t, \boldsymbol{\theta}) - \mathbf{y}^h\|^2$.

We use the STL10 dataset, and the training complexity of SGD with Multiscale-SGD from Algorithm 1, and Full-Multiscale-SGD from Algorithm 2 on several architectures. The results are presented in Table 4, with additional results on the CelebA dataset in Table 6. We notice that both

| Training Strategy | Iterations | #WU ($\downarrow$) |
|---|---|---|
| SGD (Single Scale) | 200 | 12800 |
| Multiscale-SGD (Ours) | 200 | 187 |
| Full-Multiscale-SGD (Ours) | **145** | 81 |

Table 3: Iterations and #WU for blur estimation. Multiscale training significantly reduces costs.

| Training Strategy | #WU ($\downarrow$) | MSE ($\downarrow$) | | | |
|---|---|---|---|---|---|
| | | UNet | ResNet | MFC-UNet | MFC-ResNet |
| SGD (Single Scale) | 480,000 | 0.1918 | 0.1629 | 0.1800 | 0.1862 |
| Multiscale-SGD (Ours) | 74,000 | 0.1975 | 0.1653 | 0.1719 | 0.1522 |
| Full-Multiscale-SGD (Ours) | **28,750** | 0.1567 | 0.1658 | 0.2141 | 0.1744 |

Table 4: Comparison of training strategies on the denoising task on the STL10 dataset.

multiscale training strategies achieved similar performance to SGD for all networks (see Table 4 and Table 6) with a considerably lower number of #WU . The calculations for #WU for the three training strategies have been presented in Appendix C. We provide visualizations of the results in Appendix G, and additional details on the experiment in Appendix E.

**Image Super-Resolution.** Here we aim to predict a high-resolution image $\mathbf{u}^h$ from a low-resolution image $\mathbf{y}^l$, which is typically a downsampled version of $\mathbf{u}^h$. The downsampling process is modeled as $\mathbf{y}^l = D\mathbf{u}^h + \mathbf{z}$, where $D(\cdot)$ is a downsampling operator (e.g., bicubic), and $\mathbf{z}$ is noise. The goal is to invert this process and reconstruct $\mathbf{u}^h$ using a neural network $f(\mathbf{y}^l, \boldsymbol{\theta})$. The loss function is $\text{loss}(\boldsymbol{\theta}) = \frac{1}{2}\mathbb{E}_{\mathbf{y}^l, \mathbf{u}^h} \left\| f(\mathbf{y}^l, \boldsymbol{\theta}) - \mathbf{u}^h \right\|^2$. The model $f(\mathbf{y}^l, \boldsymbol{\theta})$ predicts the high-resolution image $\mathbf{u}^h$ from the low-resolution input $\mathbf{y}^l$. In Table 5, our Full-Multiscale-SGD significantly accelerates training while maintaining image quality, measured by SSIM. Additional details and visualizations are provided in Appendix E and Appendix G.

| Training Strategy | #WU ($\downarrow$) | SSIM ($\uparrow$) | | | |
|---|---|---|---|---|---|
| | | ESPCN | ResNet | MFC-ESPCN | MFC-ResNet |
| SGD (Single Scale) | 16,000 | 0.84 | 0.82 | 0.84 | 0.83 |
| Multiscale-SGD (Ours) | 2,500 | 0.82 | 0.81 | 0.83 | 0.83 |
| Full-Multiscale-SGD (Ours) | **340** | 0.81 | 0.81 | 0.82 | 0.83 |

Table 5: Comparison of SSIM for ESPCN (Shi et al., 2016), ResNet, MFC-ESPCN and MFC-ResNet on the Urban100 dataset for the Super-Resolution using different training strategies.

# 6 CONCLUSIONS

In this paper, we introduced a novel approach to multiscale training for deep neural networks, addressing the limitations of standard mesh-based convolutions. We showed that theoretically, standard convolutions may not converge as $h \to 0$ for noisy inputs, hindering performance. To solve this, we proposed mesh-free convolutions inspired by parabolic PDEs with trainable parameters. Our experiments demonstrated that mesh-free convolutions ensure consistent, mesh-independent convergence, also in the presence of noisy inputs. Moreover, our empirical findings suggest that our multiscale training approach can also be coupled with standard convolutions – positioning our multiscale training approach as an alternative to standard SGD. This approach is well-suited for high-resolution and 3D learning tasks where scalability and precision are crucial. Although the current implementation of FFT-based convolutions like our MFCs is slower than standard convolutions, their theoretical FLOPs count is competitive. We hope that the theoretical merits discussed in this paper will inspire the development of efficient implementations of such methods.

## ETHICS AND REPRODUCIBILITY STATEMENTS

**Ethics Statement.** In this work, we do not release any datasets or models that could be misused, and we believe our research carries no direct or indirect negative societal implications. We do not work with sensitive or privacy-related data, nor do we develop methods that could be applied to harmful purposes. To the best of our knowledge, this study raises no ethical concerns or risks of negative impact. Additionally, our research does not involve human subjects or crowdsourcing. We also confirm that there are no conflicts of interest or external sponsorships influencing the objectivity or results of this study.

**Reproducibility Statement.** In Section 5, we outline the setups employed in our experiments, followed by additional details in Appendix E, such as dataset descriptions, the experimental settings for each task, and the hyperparameters used in our studies. All our experiments are conducted on publicly available datasets. We provide implementation details of our method that are reported in Section 3.1, and in Algorithm 1 and Algorithm 2. To further facilitate the reproducibility of our work, we will release all the data and code to reproduce our empirical evaluation upon acceptance.

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

## A    BACKGROUND ON THE SOLUTION OF PARABOLIC EQUATIONS

The convolution we use is defined by the solution of the parabolic PDE

$$v_t = \mathcal{L}v \qquad v(0, \mathbf{x}) = \mathbf{R}u \tag{26}$$

equipped with periodic boundary conditions and on the bounded domain $[0, 1]^2$. We now review the basic principles of parabolic PDEs to shed light on this choice.

We start with the operator $\mathcal{L}$ that has the form

$$\mathcal{L}v = \alpha_{xx}\frac{\partial^2 v}{\partial x^2} + 2\alpha_{xy}\frac{\partial^2 v}{\partial x \partial y} + \alpha_{yy}\frac{\partial^2 v}{\partial y^2} \tag{27}$$

We say that the operator is elliptic if

$$\alpha_{xx}\alpha_{yy} - \alpha_{xy}^2 > 0 \tag{28}$$

The ellipticity of the operator guarantees that it has an inverse (up to a constant) and that the parabolic Equation 27 is well-posed.

It is easy to verify that the eigenfunctions are given by Fourier vectors of the form

$$\phi_{mn}(x, y) = \exp(2\pi i(mx + ny))$$

Substituting an eigenfunction in the differential operator, we obtain that the corresponding negative eigenvalue is

$$\lambda_{mn} = 4\pi^2 \left(\alpha_{xx}m^2 + \alpha_{yy}n^2 + 2\alpha_{xy}nm\right)$$

Note that the eigenvalues are negative as long as Equation 28 holds.

Consider now a time-dependent problem given by Equation 26. Consider a solution of the form

$$v(t, x, y) = \sum_{mn} v_{mn}(t)\phi_{mn}(x, y)$$

Substituting, we obtain an infinite set of ODEs of the form

$$\frac{\partial v_{mn}}{\partial t} = -\lambda_{mn}v_{mn} \tag{29}$$

with the solution

$$v_{mn} = \exp(-\lambda_{mn}t)v_{mn}(0).$$

Since $\lambda_{mn} > 0$, the solution for each component decays to 0 with faster decay for high-frequency components. Thus, the convolution is well-posed as long as the condition in Equation 28 is kept.

## B    PROOFS

### B.1    PROOF TO LEMMA 1

*Proof.* To prove Lemma 1, we further analyze Example 1 by asking: *how does the gradient of the function behave as we downsample the mesh?* To this end, we study the linear model in Example 1 for a single input and output channels, with a 1D convolution kernel of size 3. The loss to be optimized is defined as:

$$\ell(\boldsymbol{\theta}, \mathbf{u}^h, \mathbf{y}^h) = \frac{1}{n}(\mathbf{y}^h)^\top(\boldsymbol{\theta} \star \mathbf{u}^h) = \frac{1}{n}\sum_j(\boldsymbol{\theta}_1\mathbf{u}_{j-1}^h + \boldsymbol{\theta}_2\mathbf{u}_j^h + \boldsymbol{\theta}_3\mathbf{u}_{j+1}^h)\mathbf{y}_j^h, \tag{30}$$

where $\boldsymbol{\theta}$ are trainable convolution weights, and $\mathbf{y}^h$ is a discretization of some function on the grid $\mathbf{x}^h$. The derivative with respect to $\boldsymbol{\theta}$ is given by:

$$\mathbf{g}_1^h = \frac{1}{n}\sum\mathbf{y}_j^h\mathbf{u}_{j+1}^h, \qquad \mathbf{g}_2^h = \frac{1}{n}\sum\mathbf{y}_j^h\mathbf{u}_j^h, \qquad \mathbf{g}_3^h = \frac{1}{n}\sum\mathbf{y}_j^h\mathbf{u}_{j-1}^h. \tag{31}$$

Let us also consider the same analysis, but on a nested mesh with mesh size $2h$, that yields:

$$\mathbf{g}_1^{2h} = \frac{2}{n}\sum\mathbf{y}_j\mathbf{u}_{j+2}, \qquad \mathbf{g}_2^{2h} = \frac{2}{n}\sum\mathbf{y}_j^h\mathbf{u}_j^h, \qquad \mathbf{g}_3^{2h} = \frac{2}{n}\sum\mathbf{y}_j^h\mathbf{u}_{j-2}^h. \tag{32}$$

If both functions $u(x)$ and $y(x)$ are smooth, then, we can use Taylor's theorem to obtain

$$\mathbf{y}_j\mathbf{u}_{j+2} = \mathbf{y}_j(\mathbf{u}_{j+1} + h\mathbf{u}_{j+1}' + \mathcal{O}(h^2)) = \mathbf{y}_j\mathbf{u}_{j+1} + \mathcal{O}(h),$$

and therefore, the gradients computed on mesh sizes $h$ and $2h$ are $\mathcal{O}(h)$ away, where the error depends on the magnitude of the derivative of $u(x)$ at point $\mathbf{x}_{j+1}$. $\qquad\square$

## B.2 PROOF TO LEMMA 2

*Proof.* To prove the Lemma, we recall that the convolution is invariant under Fourier transform. Indeed, let $\widehat{u}(k_x, k_y) = \mathcal{F}u$ and $\widehat{y}(k_x, k_y) = \mathcal{F}y$, then, we can write

$$\ell^*(\boldsymbol{\xi}) = \int \mathcal{L}(k_x, k_y)\widehat{u}(k_x, k_y)\widehat{y}(k_x, k_y) \, dk_x dk_y. \tag{33}$$

Since the symbol $\lim_{k_x, k_y} \mathcal{L}(k_x, k_y) \to 0$ exponentially, the loss in continuous space, $\ell^*$ converges to a finite quantity. Moreover, the derivatives of the loss with respect to its parameters are

$$\frac{\partial \ell^*}{\partial \alpha_{ml}} = -\int k_m k_l \tau \mathcal{L}(k_x, k_y)\widehat{y}(k_x, k_y)\widehat{u}(k_x, k_y) \, dk_x dk_y \quad m, l = x, y \tag{34a}$$

$$\frac{\partial \ell}{\partial \gamma_0} = \int \widehat{y}(k_x, k_y)\widehat{u}(k_x, k_y)\mathcal{L}(k_x, k_y) \, dk_x dk_y \tag{34b}$$

$$\frac{\partial \ell^*}{\partial \gamma_l} = \int \widehat{y}(k_x, k_y)\widehat{u}(k_x, k_y)ik_l\mathcal{L}(k_x, k_y) \, dk_x dk_y \ \ l = x, y \tag{34c}$$

which also converges to a finite quantity by Parseval's theorem.

$\square$

## B.3 EQUIVALENCE BETWEEN REGULAR CONVOLUTIONS AND DISCRETIZED MFCS ON A FIXED MESH

**Lemma 4.** *Given a fixed mesh of size $h$ and a $k \times k$ convolution $\mathbf{K}$. There exists a MFC $\mathbf{K}_{\boldsymbol{\theta}}$ with parameters $\boldsymbol{\theta}$ that is discretized on the same mesh such that given $\epsilon > 0$*

$$\|\mathbf{K}_{\boldsymbol{\theta}}\mathbf{u} - \mathbf{K}\mathbf{u}\|^2 \le \epsilon, \quad \forall \mathbf{u} \tag{35}$$

The implications of the lemma is that given a standard network it is always possible to replace the convolutions with MFC and remain within the same accuracy, as our experiments show.

*Proof.* The proof for this lemma is straight forward. Notice that the convolution is made from two parts. The first is a spatial component that involves the Fourier transform and the operation in Fourier space and the second is a $1 \times 1$ convolution that combines the spacial channels. We prove the lemma for a single channel. For problems with $c$ channels it is possible to concatenate the same structure $c$ times with different parameters.

To obtain any $k \times k$ convolution we need to be able to express the convolution stencil by a basis of $k^2$ convolutions that are linearly independent. To be specific, the standard convolution can be written as

$$\mathbf{K}\mathbf{u} = \sum_{j=1}^{k}\sum_{i=1}^{k} \mathbf{K}_{ij}(\mathbf{e}_{ij} \star \mathbf{u}) \tag{36}$$

where $\mathbf{e}_{ij}$ represents a convolution with a stencil of zeros and 1 in the $i, j$ location and $\mathbf{K}_{ij}$ is the $ij$ entry of $\mathbf{K}$. For Mesh Free convolution discretize on a regular mesh and using different values of $\alpha_{xy}$ then the solutions of the parabolic PDE are linearly independent (Evans, 1998). Now use any combination the parameters $\boldsymbol{\gamma}$ and vary $\alpha_{xy}$ over $m \ge k^2$ different values. Choose $\tau$ sufficiently small such that it covers the same apparatus of the standard convolution. We obtain $k^2$ different spatial convolutions $\widehat{\mathbf{e}}_i$, each corresponds to a different tilting parameter $\alpha_{xy}$. Since the new set of convolutions are linearly independent they form a basis for any $m$ convolution. In particular, it is possible to minimize the over-determined linear system for the coefficients $\mathbf{c}$ such that

$$\min_{\mathbf{c}} \left\| \sum_{j=1}^{k}\sum_{i=1}^{k} \mathbf{K}_{ij}\mathbf{e}_{ij} - \sum_{i=1}^{m} \mathbf{c}_i\widehat{\mathbf{e}}_i(\boldsymbol{\theta}) \right\|^2 \tag{37}$$

$\square$

## C COMPUTATION OF #WU WITHIN A MULTISCALE FRAMEWORK

**Definition 1** (Working Unit (WU)). *A single working unit (WU) is defined by the computation of a model (neural network) on an input image on its original (i.e., highest) resolution.*

**Remark 1.** *To measure the number of working units (#WUs) required by a neural network and its training strategy, we measure how many evaluations of the highest resolution are required. That is, evaluations at lower resolutions are weighted by the corresponding downsampling factors. In what follows, we elaborate on how #WUs are measured.*

We now show how to measure the computational complexity in terms of #WU for the three training strategies SGD, Multiscale-SGD, and Full-Multiscale-SGD. For the Multiscale-SGD approach, all computations happen on the finest mesh (size $h$), while for Multiscale-SGD and Full-Multiscale-SGD, the computations are performed at 4 resolutions $(h, 2h, 4h, 8h)$. The computation of running the network on half resolution is 1/4 of the cost, and every coarsening step reduces the work by an additional factor of 4. In the Equations below, we denote use the acronyms *MSGD* for Multiscale-SGD, and *FMSGD* for Full-Multiscale-SGD From equation 6, we have,

$$\mathbb{E}\mathbf{g}^h(\boldsymbol{\theta}) = \mathbb{E}\left[\mathbf{g}^h(\boldsymbol{\theta}) - \mathbf{g}^{2h}(\boldsymbol{\theta})\right] + \mathbb{E}\left[\mathbf{g}^{2h}(\boldsymbol{\theta}) - \mathbf{g}^{4h}(\boldsymbol{\theta})\right] + \mathbb{E}\left[\mathbf{g}^{4h}(\boldsymbol{\theta}) - \mathbf{g}^{8h}(\boldsymbol{\theta})\right] + \mathbb{E}\left[\mathbf{g}^{8h}(\boldsymbol{\theta})\right] \quad (38)$$

With Multiscale-SGD, the number of #WU in one iteration needed to compute the $\mathbb{E}\mathbf{g}^h(\boldsymbol{\theta})$ is given by,

$$\text{\#WU}_{\text{MSGD}} = N_0\left(1 + \frac{1}{4}\right) + N_1\left(\frac{1}{4} + \frac{1}{16}\right) + N_2\left(\frac{1}{16} + \frac{1}{64}\right) + \frac{N_3}{64} \quad (39)$$

where $N_0, N_1, N_2$ and $N_3$ represent the batch size at different scales. With $N_1 = 2N_0$, $N_2 = 4N_0$ and $N_3 = 8N_0$, #WU for $I$ iterations become,

$$\text{\#WU}_{\text{MSGD}} = \frac{37N_0 I}{16} \approx 2.31 N_0 I \quad (40)$$

Alternately, seeing an equivalent amount of data, doing these same computations on the finest mesh with SGD, the #WU per iteration is given by, $N_0 \times 1 + N_1 \times 1 + N_2 \times 1 + N_3 \times 1$ images in one iteration (where each term is computed at the finest scale). With $N_1 = 2N_0$, $N_2 = 4N_0$, and $N_3 = 8N_0$, the total #WU for $I$ iterations in this case, becomes,

$$\text{\#WU}_{\text{SGD}} = 15N_0 I \quad (41)$$

Thus, using Multigrid-SGD is roughly $6.5$ times cheaper than SGD.

The computation of #WU for the Full-Multiscale-SGD framework is more involved due to its cycle taking place at each level. As a result, #WU at resolutions $h, 2h, 4h$ and $8h$ can be computed as,

$$\text{\#WU}_{\text{FMSGD}}(h) = I_h \times \left[N_0^h\left(1 + \frac{1}{4}\right) + N_1^h\left(\frac{1}{4} + \frac{1}{16}\right) + N_2^h\left(\frac{1}{16} + \frac{1}{64}\right) + \frac{N_3^h}{64}\right] \quad (42)$$

$$\text{\#WU}_{\text{FMSGD}}(2h) = \frac{I_{2h}}{4} \times \left[N_0^{2h}\left(1 + \frac{1}{4}\right) + N_1^{2h}\left(\frac{1}{4} + \frac{1}{16}\right) + \frac{N_2^{2h}}{16}\right] \quad (43)$$

$$\text{\#WU}_{\text{FMSGD}}(4h) = \frac{I_{4h}}{16} \times \left[N_0^{4h}\left(1 + \frac{1}{4}\right) + \frac{N_1^{4h}}{4}\right] \quad (44)$$

$$\text{\#WU}_{\text{FMSGD}}(8h) = \frac{I_{8h}}{64} \times N_0^{8h} \quad (45)$$

$$\quad (46)$$

where $I_h, I_{2h}, I_{4h}$ and $I_{8h}$ represent the number of training iterations at each scale. Choosing $I$ iterations at the coarsest scale with $I = I_{8h} = 2I_{4h} = 4I_{8h} = 8I_h$ and $N_1^r = 2N_0^r$, $N_2^r = 4N_0^r$, and $N_3^r = 8N_0^r$ for each $r \in \{h, 2h, 4h, 8h\}$, the total #WU for Full-Multiscale-SGD become,

$$\text{\#WU}_{\text{FMSGD}} = \frac{37}{16 \cdot 8}N_0^h I + \frac{17}{32 \cdot 4}N_0^{2h} I + \frac{7}{64 \cdot 2}N_0^{4h} I + \frac{1}{64}N_0^{8h} I \quad (47)$$

Finally, choosing $N_0^{2^j h} = 2^j N_0$, total #WU simplifies to,

$$\text{\#WU}_{\text{FMSGD}} = \frac{115}{128} N_0 I \approx 0.90 N_0 I \tag{48}$$

Thus, it is roughly 16 times more effective than using SGD.

# D ADDITIONAL RESULTS

## D.1 EXPERIMENTS FOR THE DENOISING TASK ON THE CELEBA DATASET

To observe the behavior of the Full Multiscale algorithm, we performed additional experiments for the denoising task on the CelebA dataset using UNet, ResNet, MFC-UNet, and MFC-ResNet. The results have been presented in Table 6 showing that both multiscale training strategies achieved similar performance to SGD for all networks with a considerably lower number of #WU .

| Training Strategy | #WU ($\downarrow$) | MSE ($\downarrow$) | | | |
|---|---|---|---|---|---|
| | | UNet | ResNet | MFC-UNet | MFC-ResNet |
| SGD (Single Scale) | 480,000 | 0.0663 | 0.0553 | 0.0692 | 0.0699 |
| Multiscale-SGD (Ours) | 74,000 | 0.0721 | 0.0589 | 0.0691 | 0.0732 |
| Full-Multiscale-SGD (Ours) | **28,750** | 0.0484 | 0.0556 | 0.0554 | 0.0533 |

Table 6: Comparison between the performance of different networks (UNet, ResNet, MFC-UNet, and MFC-ResNet) for the denoising task on the CelebA dataset using various training strategies. We report the mean MSE computed over 512 images from the validation set and the total #WU under each training framework.

## D.2 EXPERIMENTS AND COMPARISONS WITH DEEPER NETWORKS

While we have shown that MFCs can replace convolutions on shallow networks as well as UNets, in our following experiment, we show that they can perform well even for deeper networks. To this end, we compare the training to ResNet-18 and ResNet-50, as well as UNets with 5 levels, for the super-resolution task on Urban 100. The results are presented in Table 7.

## D.3 COMPARISON WITH FOURIER BASE CONVOLUTIONS IN SUPER-RESOLUTION TASKS

We conducted experiments on the Urban100 dataset to evaluate the performance of different convolutional techniques in super-resolution tasks. Specifically, we compared the widely used ESPCN architecture with our proposed MFC-ESPCN (which uses parabolic, mesh-free convolutions) and FNO-ESPCN (which incorporates spectral convolutions used by the FNO paper). The results, measured using the SSIM metric, highlight that our MFC-ESPCN achieves comparable or better performance than standard convolutional methods while requiring lower computational resources. Furthermore, MFC-ESPCN outperforms the FNO-based spectral convolution approach, as it effectively captures both local and global dependencies without being limited by the truncation of frequencies in the Fourier domain. The results for the task of multi-resolution are presented in Table 8

## D.4 EXPERIMENTS WITH A FIXED BUDGET

An additional way to observe convergence is to plot the convergence as a function of work units. In this way, it is possible to assess the convergence of different methods for giving some value to the work units. To this end, we plot such a curve in Figure 5.

## D.5 INCORPORATION OF MFCS IN SEGMENT-ANYTHING MODEL

While mesh-free convolutions are designed to have an adaptive receptive field that can change for different problems, they can also be used as a replacement in standard architectures. To this end, we

| Training Strategy | #WU ($\downarrow$) | MSE ($\downarrow$) | | | | | |
|---|---|---|---|---|---|---|---|
| | | ResNet18 | ResNet50 | MFC-ResNet18 | MFC-ResNet50 | UNet(5 levels) | MFC-Unet(5 levels) |
| SGD (Single Scale) | 480,000 | 0.1623 | 0.1614 | 0.1620 | 0.1611 | 0.1610 | 0.1609 |
| Multiscale-SGD (Ours) | 74,000 | 0.1622 | 0.1611 | 0.1617 | 0.1602 | 0.1604 | 0.1605 |
| Full-Multiscale-SGD (Ours) | **28,750** | 0.1588 | 0.1598 | 0.1598 | 0.1599 | 0.1597 | 0.1594 |

Table 7: Comparison of training strategies on the denoising task on the STL10 dataset for ResNet18, ResNet50 and a deeper UNets.

| Training Strategy | SSIM ($\uparrow$) | | |
|---|---|---|---|
| | ESPCN | MFC-ESPCN | FNO-ESPCN |
| SGD (Single Scale) | 0.841 | **0.842** | 0.744 |
| Multiscale-SGD (Ours) | 0.822 | **0.831** | 0.743 |
| Full Multiscale-SGD (Ours) | 0.811 | **0.823** | 0.732 |

Table 8: Performance Comparison of ESPCN, MFC-ESPCN, and FNO-ESPCN across different training strategies, on the super-resolution task on the Urban100 dataset. SSIM is used as the evaluation metric, where higher values indicate better image similarity to the ground truth.

conducted the following experiment. We downloaded the popular *Segment Anything* model (SAM) with ViT-B backbone, trained on 11 million images from the SA-1B dataset (Kirillov et al., 2023). This network contains a number of standard 2D convolutions. We replaced all 2D convolutions with kernel size 3×3 with our MFCs, and trained only the parameters for MFCs, keeping the rest of the model weights frozen during training. The training was performed on a single image obtained from the SA-1B dataset (Kirillov et al., 2023). The final trained model, MFC-SAM, was tested on 1000 images from validation of the MS COCO (Lin et al., 2014) and ADE20K (Zhou et al., 2019) segmentation datasets. The inference was performed using the `SamAutomaticMaskGenerator` (Kirillov et al., 2023) with `points_per_side = 64`. We present the results comparing the performance of SAM and MFC-SAM on mean predicted IoU and mean stability score metrics in Table 9, which shows the very similar performance of SAM and MFC-SAM in terms of the quality of the predicted segmentation masks. A sample comparison between segmentation is shown in Figure 6.

Given Lemma 4, the results of replacing 2D convolutions with MFCs are not surprising as it is always possible to replace a standard convolution with an MFC on a particular mesh.

| Data | Methods | Mean predicted IoU ($\uparrow$) | Mean stability score ($\uparrow$) |
|---|---|---|---|
| COCO | SAM | 0.934 | 0.968 |
| | MFC-SAM | 0.928 | 0.965 |
| ADE20k | SAM | 0.931 | 0.969 |
| | MFC-SAM | 0.928 | 0.966 |

Table 9: Comparison between SAM and MFC-SAM for the prediction of segmentation masks on mask quality metrics like predicted IoU and stability score on 1000 images from the MS COCO and ADE20K datasets.

### D.6 TRAINING FOR LONGER

We have experimented with longer training of our network by doubling the number of epochs of the experiment reported in Table 7 for the ResNet-18 and ResNet-50. The results for a single mesh (SGD) are summarized in Table 10.

### D.7 COMPARISON OF RUNTIMES BETWEEN A SINGLE STANDARD 2D CONVOLUTION AND MFC

We compared the runtimes of a single standard 2D convolution (`nn.Conv2d` from PyTorch) over different kernel sizes ($3 \times 3$, $5 \times 5$, $7 \times 7$, $9 \times 9$ and $13 \times 13$) and image sizes ranging from $32 \times 32$

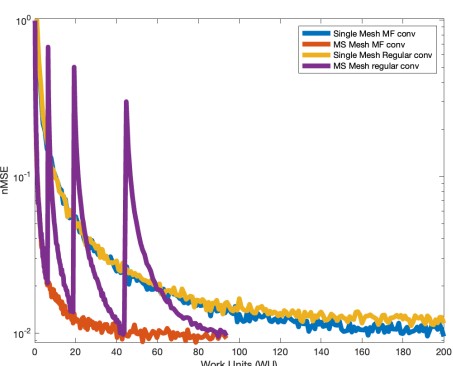

Figure 5: Convergence of different methods as a function of work-units

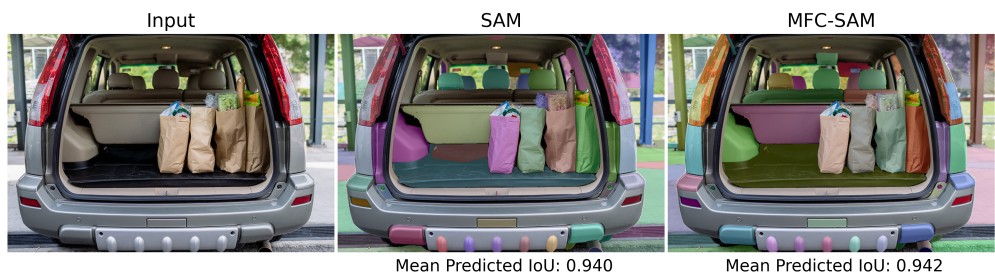

Figure 6: **Left:** The input image; **Middle:** Predicted masks using SAM (ViT-B backbone); **Right:** Predicted masks using MFC-SAM, where all $3 \times 3$ convolutions in SAM are replaced by our proposed MFCs.

to $512 \times 512$ with 3 channels. Similarly, we also computed runtimes for a single MFC operation (with scaling factor $\tau = 0.1$) over different image sizes. All operations were performed on an Intel(R) Xeon(R) Gold 5317 CPU @ 3.00GHz with x86_64 processor, 48 cores, and a total available RAM of 819 GB. The average of the runtimes (in milliseconds) over 500 operations for all experiments have been presented in Figure 7. In addition, we report the FLOPs required by standard convolutions with varying kernel and input image sizes, with the FLOPs required by our MFCs, in Table 11.

### D.8 MULTISCALE-SGD VS. RANDOM CROPS

A method that implicitly includes multiscale training is that of the data-augmentation technique of random crops. We now compare the performance between using random crops and our Multiscale-SGD, using the *'RandomResizedCrop'* transform from torchvision in PyTorch (Paszke et al., 2017). As shown in Table 12, the performance with random crops exhibits a slight deterioration compared to fixed-size images. This reduction in performance can likely be attributed to the smaller receptive field introduced by the use of random crops, compared with our Multiscale-SGD, which uses multiple resolutions of the image.

### D.9 RESULTS WITH DIFFUSION MODELS

We now study the applicability of our Multiscale training framework to the training of a diffusion model. Specifically, we use the architecture and training loss in Denoising Diffusion Probabilistic Models (DDPM) (Ho et al., 2020) coupled with our Full-Multiscale-SGD framework, on the MNIST dataset (Lecun & Cortes). We compare the generated images by the standard DDPM and DDPM augmented with our Full-Multiscale-SGD framework in Figure 8, alongside the ground-truth images from MNIST, for reference. As evident, both methods yield images that are similar to MNIST images, indicating the applicability of our Multiscale training framework for diffusion models.

| Double #iterations | Architecture | MSE ($\downarrow$) |
|---|---|---|
| No | ResNet-18 | 0.1622 |
| No | ResNet-50 | 0.1611 |
| Yes | ResNet-18 | 0.1599 |
| Yes | ResNet-50 | 0.1585 |

Table 10: Experiments with longer training (double the number of iterations) of ResNet18 and ResNet50 on the denoising task on the STL10 dataset.

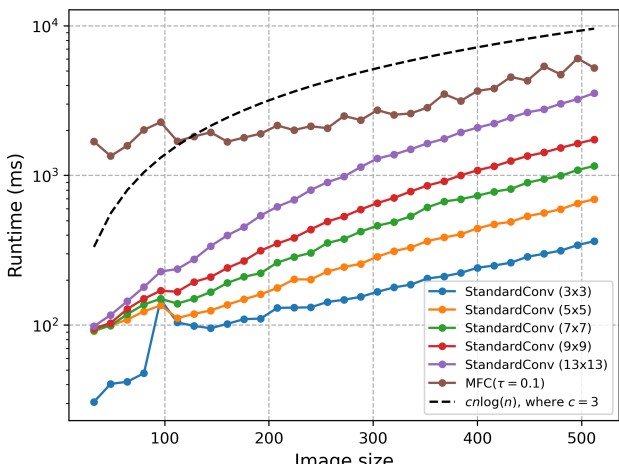

Figure 7: Comparison of runtimes (milliseconds) for a single standard 2D convolution with different kernel sizes ($3 \times 3$, $5 \times 5$, $7 \times 7$, $9 \times 9$ and $13 \times 13$) and MFC over images of different sizes ranging from $32 \times 32$ to $512 \times 512$ with 3 channels. All runs were performed on the CPU.

## E    EXPERIMENTAL SETTING

For the denoising task, the experiments were conducted on STL10 and CelebA datasets. The networks used were UNet, ResNet with standard convolutions MFC-UNet, and MFC-ResNet with mesh-free convolutions. The key details of the experimental setup are summarized in Table 13. For the image super-resolution task, the experiments were conducted on the Urban100 dataset, employing a deep residual network and utilizing a multiscale training strategy. The key details of the experimental setup, including the optimizer, loss function, and training strategies, are summarized in Table 14. All our experiments were conducted on an NVIDIA A6000 GPU with 48GB of memory. Upon acceptance, we will release our source code, implemented in PyTorch (Paszke et al., 2017).

## F    RUNTIMES

To have a fair time comparison, we measure the time for the network to process the same number of examples in each of the experiments. Let us consider the case of 4 levels within a Multiscale-SGD training strategy. In this case, a multiscale computation of the gradient processes for every single image on the finest resolution, 2 images on the next resolution, 4 on the third resolution, and 8 on the coarsest resolution. Thus, we compare its runtimes with the time it takes to process a batch of 15 fine-scale images, as with the standard approach of SGD. The results are measured on ResNet and MFC-ResNet architectures.

We observe a significant time reduction when considering the multiscale strategy. Note that the MFC-ResNet is more expensive than the standard ResNet. This is due to the computation of the Fourier transform, which is more expensive than a simple convolution.

| Image Size (n×n) / Kernel Size (k×k) | 32 | 64 | 128 | 256 |
|---|---|---|---|---|
| 5 | $2.62 \times 10^7/5.56 \times 10^7$ | $1.05 \times 10^8/2.65 \times 10^8$ | $4.19 \times 10^8/1.23 \times 10^9$ | $1.68 \times 10^9/5.57 \times 10^9$ |
| 7 | $5.14 \times 10^7/5.56 \times 10^7$ | $2.06 \times 10^8/2.65 \times 10^8$ | $8.22 \times 10^8/1.23 \times 10^9$ | $3.29 \times 10^9/5.57 \times 10^9$ |
| 9 | $8.49 \times 10^7/5.56 \times 10^7$ | $3.40 \times 10^8/2.65 \times 10^8$ | $1.36 \times 10^9/1.23 \times 10^9$ | $5.44 \times 10^9/5.57 \times 10^9$ |
| 11 | $1.27 \times 10^8/5.56 \times 10^7$ | $5.08 \times 10^8/2.65 \times 10^8$ | $2.03 \times 10^9/1.23 \times 10^9$ | $8.12 \times 10^9/5.57 \times 10^9$ |
| 13 | $1.77 \times 10^8/5.56 \times 10^7$ | $7.09 \times 10^8/2.65 \times 10^8$ | $2.84 \times 10^9/1.23 \times 10^9$ | $1.13 \times 10^{10}/5.57 \times 10^9$ |
| 15 | $2.36 \times 10^8/5.56 \times 10^7$ | $9.44 \times 10^8/2.65 \times 10^8$ | $3.77 \times 10^9/1.23 \times 10^9$ | $1.51 \times 10^{10}/5.57 \times 10^9$ |

Table 11: Comparison of FLOPs for Standard Convolutions and MFCs across different image and kernel sizes. The reported FLOPs are in the format of Standard Convolutions / our MFCs.

| Training Strategy | UNet MFC (Random crops) | UNet Regular Convs. (Random crops) | UNet MFC | UNet Regular Convs. |
|---|---|---|---|---|
| Multiscale | 0.1623 | 0.1612 | 0.1605 | 0.1604 |
| Full Multiscale | 0.1619 | 0.1601 | 0.1609 | 0.1597 |
| Single Mesh | 0.1602 | 0.1621 | 0.1594 | 0.1610 |

Table 12: Comparison of training strategies with UNet using MFCs and regular convolutions, with random crops vs. our Multiscale-SGD.

## G  VISUALIZATIONS

In this section, we include visualization of the outputs obtained from baseline UNet and ResNet models, as well as our MFC-UNet and MFC-ResNet networks. The visualizations for the super-resolution task are provided in Figure 13 and Figure 14. The visualizations for the denoising task on the STL-10 dataset are provided in Figure 9 and Figure 10. The visualizations for the denoising task on the CelebA dataset are provided in Figure 11 and Figure 12.

## H  COMPARING FEATURE MAPS BETWEEN CNNS AND MFCS

As seen in the experimental results, standard convolutions can benefit from the multiscale framework even though they may not converge in theory. We hypothesize that this is due to the smoothing properties of the filters that are obtained in the training of CNNs. To this end, we use the denoising experiment and plot the feature maps after the first layer in a ResNet-18. A few of the feature maps for our convolutions, as well as regular convolutions, are presented in Figure 15. As can be seen in the figure, both our filters and standard CNNs tend to smooth the noisy data and generate smoother feature maps. This can account for the success of standard CNN training in this experiment.

## I  ANALYSIS OF THE COST OF TRAINING COMPARED TO LOSS

One possible way to observe the advantage of multiscale training is to record the loss as a function of computational budget; that is, rather than recording the number of iterations, we record the loss as a function of computational cost. To this end we add a figure of the loss as a function of WU.

Table 13: Experimental details for training for the denoising task

| Component | Details |
|---|---|
| Dataset | STL10 and CelebA. Images from both datasets were resized to a dimension of $64 \times 64$ |
| Network architectures | UNet: 3-layer network with 32, 64, 128 filters, and 1 residual block (res-block) per layer ResNet: 2-layer residual network 128 hidden channels MFC-UNet: 3-layer network with MFCs with 32 hidden channels and 1 res-block per layer MFC-ResNet: 2-layer residual network with MFCs with 128 hidden channels |
| Number of training parameters | UNet: 2,537,187; ResNet: 597,699, MFC-UNet: 2,307,411, MFC-ResNet: 532,163 |
| Training strategies | SGD, Multiscale-SGD and Full-Multiscale-SGD for all networks |
| Loss function | MSE loss |
| Optimizer | Adam (Kingma & Ba, 2014) |
| Learning rate | $5 \times 10^{-4}$ (constant) |
| Batch size strategy | Dynamic batch sizing is used, adjusting the batch size upwards during different stages of training for improved efficiency. For details, see Appendix C. |
| Multiscale levels | 4 |
| Iterations per level | SGD and Multiscale-SGD: [2000, 2000, 2000, 2000], Full-Multiscale-SGD: [2000, 1000, 500, 250] for the 4 levels |
| Evaluation metrics | MSE over the 512 images from the validation set |

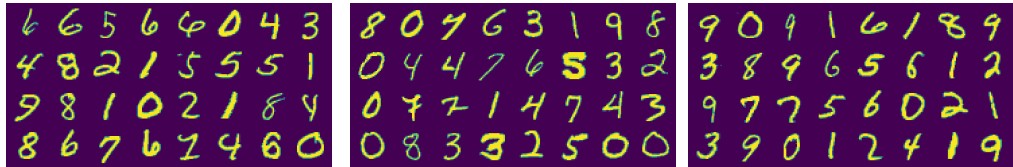

Figure 8: **Left:** Ground-truth images from MNIST; **Middle:** Images generated using DDPM with standard convolutions; **Right:** Images generated using DDPM with Full-Multiscale-SGD.

Table 14: Experimental details for training for the super-resolution task

| Component | Details |
|---|---|
| Dataset | Urban100, consisting of paired low-resolution and high-resolution image patches extracted for training and validation. |
| Network Architecture | A deep residual network (ResNet) with 5 residual blocks, specifically designed for image super-resolution. The ESPCN and the modified architectures with MFC were also included for a better comparison. |
| Training Strategy | A multiscale training approach is employed, utilizing Full-Multiscale-SGD cycles to progressively refine image resolution during training. |
| Loss Function | The Structural Similarity Index Measure (SSIM) is used to maximize perceptual quality between predicted and target images. |
| Optimizer | The Adam (Kingma & Ba, 2014) optimizer, initialized with a learning rate of 1e-3, is used for its adaptive gradient handling. |
| Learning Rate | Set to an initial value of 1e-3. |
| Batch Size Strategy | Dynamic batch sizing is used, adjusting the batch size upwards during different stages of training for improved efficiency. For details, see Appendix C. |
| Multiscale Levels | A hyperparameter capped at 4. |
| Early Stopping | Implemented based on the validation loss and gradient norms, aiming to prevent overfitting and reduce computation time. |
| Evaluation Metric | Validation loss is calculated using SSIM to evaluate the quality of the generated high-resolution images. |

| **ResNet** | | |
|---|---|---|
| Training Strategy | SGD | Multiscale-SGD (Ours, 4 levels) |
| Time (ms) | 21.5 | 1.43 |
| **MFC-ResNet** | | |
| Training Strategy | SGD | Multiscale-SGD (Ours, 4 levels) |
| Time (ms) | 27.3 | 1.79 |

Table 15: Runtime (milliseconds) per training iteration using standard convolution kernels, and our MFCs. The results are measured on ResNet and MFC-ResNet architectures with an input image size of $256 \times 256$ on an NVIDIA A6000 GPU.

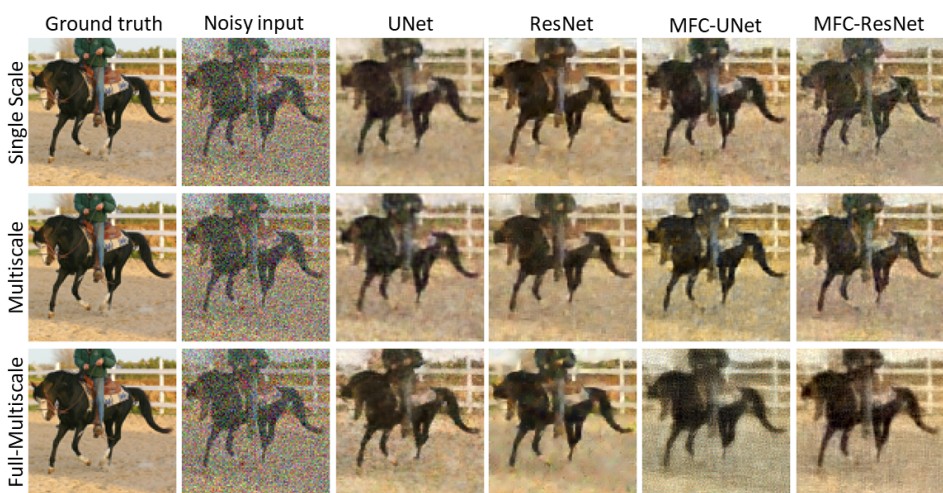

Figure 9: A comparison of different network predictions for Single Scale, Multiscale, and Full-Multiscale for an image from the STL10 dataset. The first two columns display the original image and data (same for all rows), followed by results from UNet, ResNet, MFC-UNet, and MFC-ResNet.

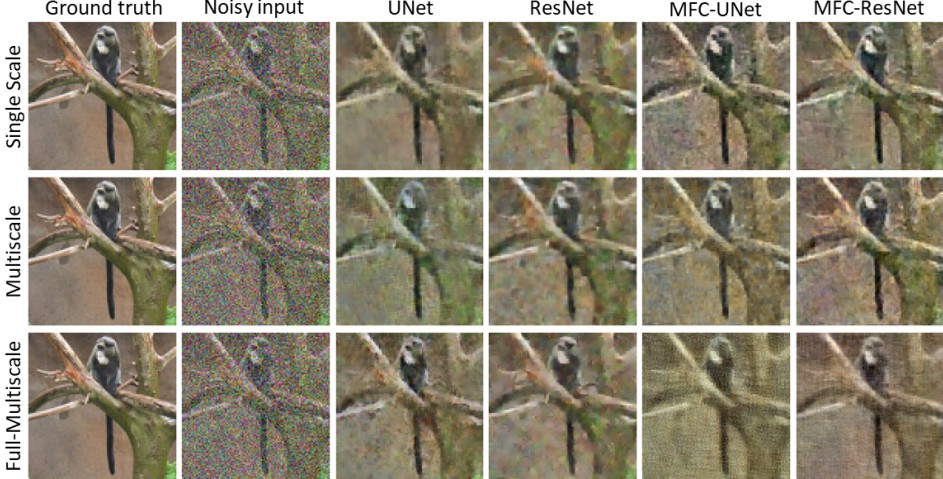

Figure 10: A comparison of different network predictions for Single Scale, Multiscale, and Full-Multiscale for an image from the STL10 dataset. The first two columns display the original image and data (same for all rows), followed by results from UNet, ResNet, MFC-UNet, and MFC-ResNet.

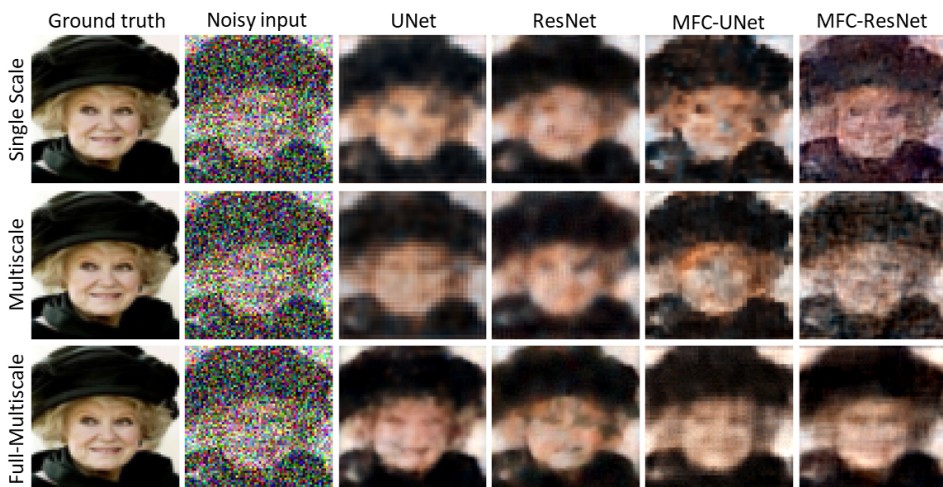

Figure 11: A comparison of different network predictions for Single Scale, Multiscale, and Full-Multiscale for an image from the CelebA dataset. The first two columns display the original image and data (same for all rows), followed by results from UNet, ResNet, MFC-UNet, and MFC-ResNet.

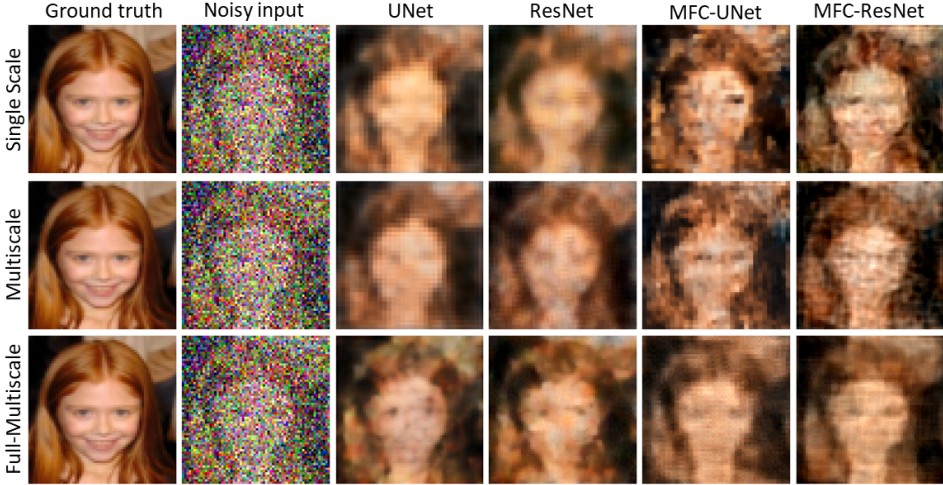

Figure 12: A comparison of different network predictions for Single Scale, Multiscale, and Full-Multiscale for an image from the CelebA dataset. The first two columns display the original image and data (same for all rows), followed by results from UNet, ResNet, MFC-UNet, and MFC-ResNet.

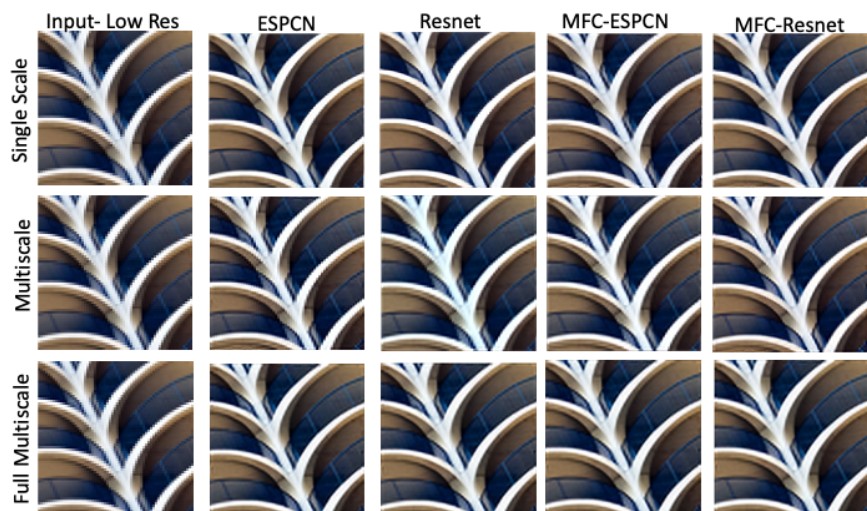

Figure 13: A comparison of different networks predictions for SGD, Multiscale-SGD, and Full-Multiscale-SGD for an image from the Urban100 dataset.

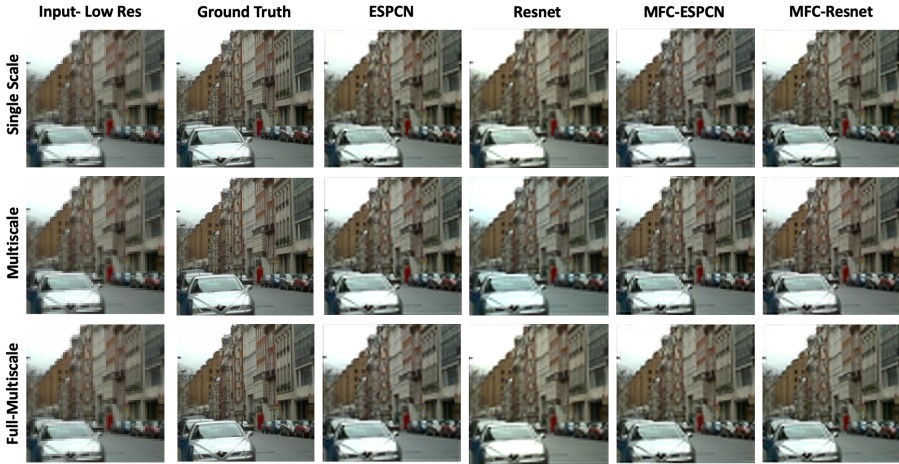

Figure 14: A comparison of different networks predictions for Single Scale, Multiscale, and Full-Multiscale for an image from the Urban100 dataset..

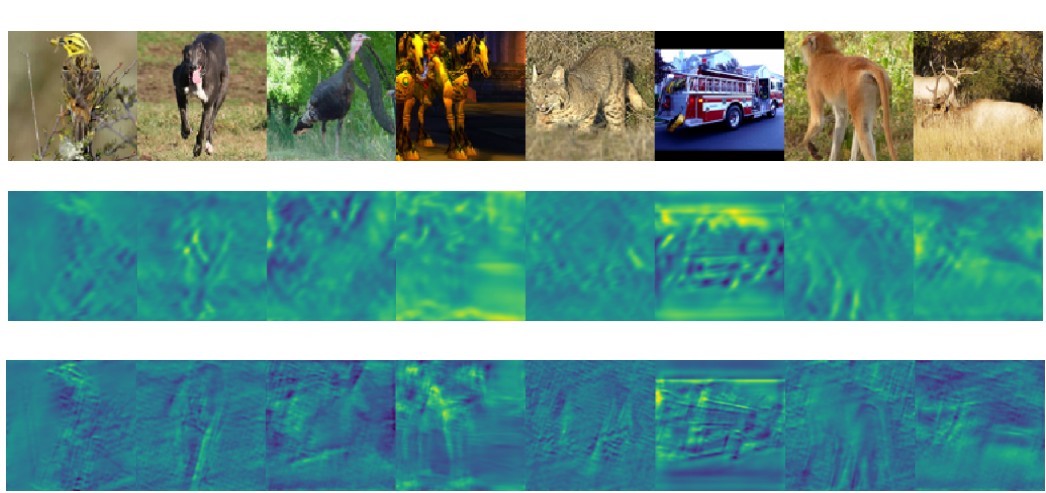

Figure 15: **First row**: Input images. **Second row**: feature maps obtained after the first layer of a ResNet-18 with our MFCs. **Third row**: feature maps obtained after the first layer of a ResNet-18 with standard convolutions. Both networks are trained on the denoising task on the STL10 dataset.

