# OpenReview forum: "Multiscale Training of Convolutional Neural Networks"
_ICLR.cc/2025/Conference — Submitted to ICLR 2025_

### Official Review · Reviewer_5Ub1 · 2024-11-01

**Soundness:** 3
**Presentation:** 3
**Contribution:** 3
**Rating:** 6
**Confidence:** 3

**Summary:**

The paper presents a training framework designed to optimize CNNs more efficiently by employing multiscale training strategies. This approach targets the significant computational challenges that arise when training CNNs at high resolutions, especially when handling noisy input data. By introducing Multiscale-SGD and the MFCs, the framework reduces computational costs and ensures stable gradient behavior across varying scales.

**Strengths:**

1.	The paper presents a good framework that integrates multiscale training with convolutional neural networks (CNNs).
2.	The writing of this paper is easy to follow.
3.	The experiments validate the effectiveness of the proposed methods, showcasing practical improvements of the proposed method.
4.	This paper offers thorough mathematical analysis, including proofs and lemmas.

**Weaknesses:**

1.	A significant limitation is the evaluation performed with shallow CNNs, all models containing fewer than four layers. Given that practical applications often leverage deeper architectures like ResNet with over 18 layers, it raises concerns about the scalability and real-world applicability of the proposed multiscale framework. Demonstrating results with deeper networks could help validate its utility for larger and more complex tasks.
2.	In the context of modern architectures, transformers such as Vision Transformers (ViT) are becoming more prominent for various applications. The paper does not explore whether the proposed method could be adapted or extended to transformer-based architectures. This is particularly relevant as transformers may handle multiscale data differently from CNNs. Discussing or experimenting with how the proposed method might extend to these architectures would strengthen the paper's broader applicability
3.	While the paper presents experimental validation of the proposed method's effectiveness, the experiments are mostly focused on simpler tasks. The applicability of the method to more complex and challenging applications, like object detection, video understanding tasks, remains untested.
4.	While the paper claims significant computational advantages with the multiscale approach, it lacks a thorough comparative analysis of computational costs with existing training methods. For instance, how this method compares in terms of hardware resource use, memory consumption, or training speed against some techniques like mixed-precision training is not clearly outlined.
5.	The paper could benefit from a deeper exploration and further discussion of how the training dynamics change with the proposed method compared to standard methods.

**Questions:**

See the above weaknesses

---

> ### Author Response · Authors · 2024-11-19
> **Rebuttal Part 1**
>
> We thank the Reviewer for recognizing the significance of our contributions, including the integration of multiscale training strategies with convolutional neural networks, the introduction of Multiscale-SGD and Mesh-Free Convolutions (MFCs), and the thorough mathematical analysis supporting our framework. We are grateful for the detailed and constructive feedback, which we address point by point in our responses below. We hope that our revisions adequately address your concerns and that you will consider revising your score.
>
> **Regarding W1 (Depth of networks):** Thank you for the suggestion. In our revised paper, we have included results using deeper networks, such as ResNet-18 and ResNet-50, on image denoising. Additionally, as recommended by Reviewer TN3W, we conducted experiments using the *Segment Anything* architecture [R1] on a semantic segmentation task, tested on the COCO and ADE-20K datasets. The results demonstrate that our MFCs achieve comparable performance to the baseline networks while significantly reducing computational costs in terms of #WUs.  We believe these additional results further strengthen the contributions of our work and enhance its relevance to the broader machine-learning community. All results are reported in our general response to all reviewers and have been added to and discussed in the revised paper in Appendix D. We appreciate your valuable suggestion, which has helped us improve the quality and comprehensiveness of the paper.
>
> **Regarding W2 (Transformers):** We have included an experiment using the *Segment Anything* network [R1], which incorporates transformers. Our results show that it is possible to combine our MFCs with this architecture while maintaining its performance on the COCO and ADE20K datasets.  However, the primary focus of this work is on optimization techniques for convolutional neural networks. Further exploration into the integration of multiscale training with transformers represents an intriguing future research direction. Thank you for your suggestion, which has helped broaden the scope of our work.
>
> **Regarding W3 (Other tasks):** In our paper, we focus on the tasks of image super-resolution and image denoising. During the rebuttal period, inspired by the insightful comments and suggestions from you and the other reviewers, we expanded our experiments to include additional tasks such as image segmentation. These new experiments utilized deeper architectures, such as ResNet-50, and landmark CV models like *Segment Anything* as suggested by Reviewer TN3W.  This comprehensive suite of experiments now spans more than four datasets, four architectures, and three tasks, further demonstrating the effectiveness of our method, as recognized by the reviewer. Extending our framework to other domains, such as video understanding, is an exciting direction that we leave for future work. Thank you for your valuable feedback, which has strengthened our paper.
>
>
> **Regarding W4 (Other training methods):** Thank you for the thoughtful suggestion. In our paper, we focus on the training strategy from the perspective of optimization algorithms. As discussed in Section 1, most modern approaches rely on gradient-based methods, where gradients are computed with respect to the input provided to the network, whether it is resized or cropped during preprocessing. Our work identifies and develops a multiscale *optimization* approach that significantly reduces computational load, as demonstrated in our paper and further expanded upon in this rebuttal.
>
> We agree with the Reviewer that other tangential approaches, such as quantization or hardware-specific optimizations, exist. However, these methods are complementary rather than competing with our approach: quantization focuses on hardware representation, whereas our work addresses the optimization of network weights. Inspired by your suggestion, we have now conducted additional experiments. We have added a figure that compares the runtimes of the MFCs to standard convolutions and provided detailed flop counts for each of these techniques. These are now presented in Figure 7 and Table 11.   Thank you for your valuable feedback, which has helped us further strengthen our work.

---

> > ### Author Response · Authors · 2024-11-19
> > **Rebuttal Part 2**
> >
> > **Regarding W5 (Training behavior):** Thank you for the suggestion. Following your guidance, we have expanded Figure 4, which shows the training loss curves for standard SGD and our Multiscale-SGD, by including an additional curve that plots the training loss against the number of WUs. This updated analysis is presented in Figure 5 of our revised paper.  As evident from Figure 5, our Multiscale-SGD demonstrates faster convergence in terms of training loss compared to standard SGD. We believe this result further emphasizes the contribution and significance of our approach. Thank you for your valuable feedback, which has helped strengthen our paper.
> >
> > **References:**
> >
> > [R1]  Alexander Kirillov, Eric Mintun, Nikhila Ravi, Hanzi Mao, Chloe Rolland, Laura Gustafson, Tete Xiao, Spencer Whitehead, Alexander C. Berg, Wan-Yen Lo, Piotr Dollár, Ross Girshick, Segment Anything, arXiv:2304.02643 (2023)

---

> ### Comment · Reviewer_5Ub1 · 2024-11-23
> **Response**
>
> Thank you for the rebuttal. It's good to see that this work can be adapted to ResNet-50 and SAM. Most of my concerns have been successfully addressed, and I have increased my score accordingly.

---

### Official Review · Reviewer_TN3W · 2024-11-02

**Soundness:** 2
**Presentation:** 2
**Contribution:** 2
**Rating:** 3
**Confidence:** 3

**Summary:**

This work begins with an error analysis of the the standard training procedure for convolutional networks when training at multiple resolutions, and (using some clever re-interpretations by varying mesh size and sample size) derive new multiscale training algorithms that require fewer network evaluations at the finest mesh size. They further present mesh-free convolutions that can be thought of as limits of progressively refined mesh-dependent convolutions. Finally, experimental results are presented to compare the training algorithms and network variants against deep learning standards like SGD and U-Nets/ResNets.

**Strengths:**

1) The primary research questions in the paper are well-motivated.
2) The technical analysis seems to be sound, and the authors are able to propose interesting, non-obvious training algorithms from them (i.e., this goes far beyond "multiscale training is useful for learning convolutional kernels"). I also found the further step of mesh-free convolutions to be very interesting from a theoretical perspective.

**Weaknesses:**

My major problems with the paper pertain to the experimental results, and fall into two main categories:

1) The experiments done are on extremely small networks and datasets. There are very standard collections of computer vision experiments that could be used to show the benefits of the proposed multiscale-SGD training algorithms (see, for example, the benchmarks used for experiments in any landmark CV papers from the past few years, like CLIP or Segment Anything). I understand that compute access could be a factor here, but many of these datasets are small enough that access to even a single GPU should be sufficient to perform the necessary experiments.

2) Unclear presentation of results. It's difficult to understand the relative compute requirements of different algorithms based on the work unit representation that is used in the experiments. It would be much clearer to either present the FLOPs required per iteration or backward pass, or best of all would be to simply show the amount of time taken to train the network in each case.

3) Non-competitive baselines. In practice, multiscale training is implicitly accounted for in most deep learning training procedures via data augmentations in the form of varying-sized crops of images (see the widely-used RandomResizedCrop transform in torchvision). To me, this is the actual baseline to which we should compare to, but it is not mentioned or compared against at all in your work.

Based on these, I find it difficult to be convinced by the presented experiments that the proposed multiscale algorithms would be better to use than standard SGD on usually-sized computer vision trianing tasks.

**Questions:**

1) Why are there no experiments on CV prediction tasks like classification, segmentation, object detection, etc.? Are the proposed algorithms not expected to work well in these contexts?

2) Corresponding to the data augmentation comment above: It was not clear to me if you use data augmentations like RandomResizedCrop or similar in any of your experiments; do you? If not, would you be able to show what happens when these are added to the single-scale SGD training procedure?

3) Corresponding to the comment on presentation of results comment above: could you add (even approximate) FLOPs computations or raw timings to your experiments?

4) Are there situations in which it would be preferable to use the Multiscale-SGD algorithm that you proposed instead of the Full-Multiscale-SGD algorithm? If not, it would be better to focus more of the experiments on the Full-Multiscale algorithm, and leave the Multiscale algorithm to an ablation study.

---

> ### Author Response · Authors · 2024-11-19
> **Rebuttal Part 1**
>
> We thank the Reviewer for recognizing the novelty and significance of our contributions, including the error analysis of standard training procedures, the development of new multiscale training algorithms, and the theoretical insights into Mesh-Free Convolutions. We are grateful for the detailed and constructive feedback, which we address one by one in our responses below. We hope that you find them satisfactory, and that you will consider revising your score.
>
>
> **Regarding W1 (Networks and datasets):**  Thank you for the constructive and actionable feedback. Following your guidance, we have now added a number of experiments on additional datasets and with different architectures. **All the results are provided in the Tables below, and we also added them to the revised paper in Appendix D.**
>  In particular, added the following experiments:
>
> 1. We ran a denoising experiment on the STL10 dataset with deeper networks: ResNet-18 and ResNet-50, and deeper UNets. In addition to the efficiency of our Multiscsle-SGD in terms of #WUs, the results indicate that similar to conventional CNNs with SGD,  our Multiscale-SGD, and MFCs coupled with deeper networks, yield better results in terms of downstream performance compared to shallow networks.
> | Training Strategy        | #WU (↓)   | ResNet18 | ResNet50 | MFC-ResNet18 | MFC-ResNet50 | UNet (5 levels) | MFC-UNet (5 levels) |
> |--------------------------|------------|-----------|-----------|---------------|---------------|-----------------|----------------------|
> | SGD (Single Scale)       | 480,000    | 0.1623    | 0.1614    | 0.1620        | 0.1611        | 0.1610          | 0.1609              |
> | Multiscale-SGD (Ours)    | 74,000     | 0.1622    | 0.1611    | 0.1617        | 0.1602        | 0.1604          | 0.1605              |
> | Full-Multiscale-SGD (Ours) | **28,750** | 0.1588    | 0.1598    | 0.1598        | 0.1599        | 0.1597          | 0.1594              |
>
>
> 2. Following the suggestion of Reviewer 35KC, we added a comparison with Fourier Neural Operators (FNOs), showing that our Multiscale-SGD can be coupled with FNOs and that our MFCs + Multiscale-SGD yields better performance than FNOs for the super-resolution task.
> | Training Strategy       | ESPCN   | MFC-ESPCN (Ours) | FNO-ESPCN |
> |-------------------------|---------|------------------|-----------|
> | Single Scale            | 0.841   | 0.842            | 0.744     |
> | Multiscale-SGD (Ours)   | 0.822   | 0.831            | 0.743     |
> | Full-Multiscale-SGD (Ours) | 0.811   | 0.823            | 0.732     |
>
> 3. We used the network proposed in Segment Everything [R1] and replaced their convolutions with our MFCs followed by their training while freezing the rest of the parameters. The results with our MFCs are similar to those obtained with the network in [R1], on the COCO and ADE-20K datasets. To provide further insights, we added a visualization (Figure 6 in the revised paper) to our revised appendix. We believe that these added results further strengthen the validity and practicality of our framework. Thank you
> | Data   | Methods   | Mean Predicted IoU | Mean Stability Score |
> |--------|-----------|--------------------|-----------------------|
> | COCO   | SAM       | 0.934              | 0.968                 |
> | COCO   | MFC-SAM   | 0.928              | 0.965                 |
> | ADE20k | SAM       | 0.931              | 0.969                 |
> | ADE20k | MFC-SAM   | 0.928              | 0.966                 |
>
>
>
> *Overall, we believe that the additional experiments provided in our response are help to further highlight the merit of our work, and your suggestions helped us to improve our paper. Thank you.*

---

> > ### Author Response · Authors · 2024-11-19
> > **Rebuttal Part 2**
> >
> > **Regarding W2 (Presentation of #WUs):** We appreciate your suggestions. The use of #WU in our paper provides a standardized metric to compare the number of evaluations at the highest (original) resolution required by different training strategies (e.g., SGD or our Multiscale-SGD), independent of the specific network architecture. For instance, the FLOPs required for a UNet differ from those for a ResNet. However, by employing this metric, we focus on the training strategy itself—our paper's primary contribution—while remaining invariant to the underlying architecture. Additionally, computational time can be influenced by various implementation details. For example, parallel computation of multiscale structures can significantly reduce runtimes, and the choice of parallelization strategy can have a considerable impact. By reporting the number of work units (#WU), we ensure that our metric remains unaffected by such design choices, providing a fair and architecture-independent evaluation.
> > Nonetheless, we agree that reporting additional metrics can provide valuable insights. In response to your suggestion, we have measured the number of FLOPs required by a standard convolution compared to our mesh free convolution. The results are presented in the Table below and have also been incorporated into the revised paper.This suggests that using MFCs can have similar cost to using regular convolutions if implemented correctly.
> >
> > | Image Size (n×n) / Kernel Size (k×k) | 5               | 7               | 9               | 11              | 13              | 15              |
> > |--------------------------------------|-----------------|-----------------|-----------------|-----------------|-----------------|-----------------|
> > | 32 Standard Conv / MFC               | 2.62e+07 / 5.56e+07 | 5.14e+07 / 5.56e+07 | 8.49e+07 / 5.56e+07 | 1.27e+08 / 5.56e+07 | 1.77e+08 / 5.56e+07 | 2.36e+08 / 5.56e+07 |
> > | 64 Standard Conv / MFC               | 1.05e+08 / 2.65e+08 | 2.06e+08 / 2.65e+08 | 3.40e+08 / 2.65e+08 | 5.08e+08 / 2.65e+08 | 7.09e+08 / 2.65e+08 | 9.44e+08 / 2.65e+08 |
> > | 128 Standard Conv / MFC              | 4.19e+08 / 1.23e+09 | 8.22e+08 / 1.23e+09 | 1.36e+09 / 1.23e+09 | 2.03e+09 / 1.23e+09 | 2.84e+09 / 1.23e+09 | 3.77e+09 / 1.23e+09 |
> > | 256 Standard Conv / MFC              | 1.68e+09 / 5.57e+09 | 3.29e+09 / 5.57e+09 | 5.44e+09 / 5.57e+09 | 8.12e+09 / 5.57e+09 | 1.13e+10 / 5.57e+09 | 1.51e+10 / 5.57e+09 |
> >
> > **Regarding W3 (Multiscale training):** We agree that in standard deep learning training, random crops are often employed. Our approach can be seamlessly applied to cropped images in the same manner as to the original full-size images, and the analysis presented in the paper remains valid.  The Reviewer raises a valid point: if the cropped images are very small (e.g., \(32 \times 32\)), multiscale training may not provide significant benefits since the images are already small, and the advantages of multiscale training might be minimal.  However, it is important to highlight that for tasks requiring a large receptive field, small crop sizes can lead to underperformance, as discussed in [R2]. To address this, we have added a comment to the introduction of the revised paper to acknowledge this consideration and its implications. Thank you.
> >
> > **Regarding Q1 (Experiments)**: In the original manuscript, we focused on denoising and super-resolution tasks. The primary reason for selecting denoising is its fundamental role as the core engine in diffusion-based methods, which are critical for generative models.  Based on your request, we have added experiments on segmentation. Specifically, we utilized the *Segment Everything* model and replaced its standard convolutions with our MFCs. Our results demonstrate that MFCs achieve performance on par with regular convolutions in this context.  We thank the Reviewer for suggesting this task, as we believe it strengthens our paper and helps demonstrate the broader applicability of the methods we present to the general audience.
> >
> > **Regarding Q2 (Random image cropping)**: In our initial experiments, we interpolated the images to a fixed size. Based on the Reviewer’s comment, we have now incorporated random crops of the specified size and applied the same methodology to these cropped images. Using the results from the Table below, we observe a slight deterioration in performance. This is likely because smaller random crops result in a reduced receptive field (please refer to our discussion in the response to Q1 for further details). We added the results to the revised paper in Appendix D.8. Thank you.
> >
> > | Training Strategy   | UNet MFC (Random Crops) | UNet Regular Convs. (Random Crops) | UNet MFC | UNet Regular Convs. |
> > |-----|----|---|--|--|
> > | Multiscale    | 0.1623   | 0.1612     | 0.1605   | 0.1604               |
> > | Full Multiscale   | 0.1619      | 0.1601       | 0.1609   | 0.1597               |
> > | Single Mesh              | 0.1602         | 0.1621    | 0.1594   | 0.1610               |

---

> > > ### Author Response · Authors · 2024-11-19
> > > **Rebuttal Part 3**
> > >
> > > **Regarding Q3 (FLOPs):** We thank the Reviewer for this insightful comment. Indeed, measuring FLOPs can help provide an unbiased evaluation by accounting for differences in implementations and hardware configurations. Our computation of FLOPs is as follows:
> > >
> > > For the standard convolution, we have:
> > > flops_conv2d = kernel_size^2 x im_size^2*in_c*out_c
> > >
> > > For the MFC convolutions, we have :
> > >
> > > flops_conv2d = 5*(im_size^2)*log2(im_size^2)*out_c * 2 +   im_size^2*in_c*out_c     +2*out_channels^2 *N
> > >
> > > In our response to W2, we provide a table detailing the FLOPs for standard convolutions and MFCs across various mesh sizes and kernel sizes. As shown, even with a mesh size of $64^2$, our convolution remains competitive with a  $7 \times 7$ kernel.  While our method is highly efficient in terms of FLOPs in many cases, differences in software implementation can have a significant impact on runtime. For instance, although the FLOP count breaks even for a  $7 \times 7$ convolution with a $64^2$ grid, the actual CPU or GPU runtime may favor standard convolutions due to their extensive optimization in both software and hardware, compared to FFT-based operations.
> > > We hope that our work, alongside others leveraging FFT-based methods such as FNOs [R3], will encourage GPU manufacturers to optimize their hardware to align implementation efficiencies with theoretical advantages.
> > >
> > > **Regarding Q4 (Multiscale-SGD vs. Full-Multiscale-SGD)**:  Thank you for the question. It is important to note that Full-Multiscale and Multiscale-SGD are *complementary* algorithms. Specifically, while Multiscale-SGD reduces the variance of gradients (as discussed in Section 3), the Full-Multiscale algorithm leverages a coarse mesh to provide a better initial point for the fine-mesh convolution, a technique sometimes referred to as "mesh continuation" [R4].
> > > The choice between these algorithms often depends on the smoothness of the data, as certain scenarios may favor one approach over the other. Additionally, since these mechanisms address distinct aspects of the training process, we believe that it is crucial to evaluate the effect of each algorithm in isolation.
> > >
> > > **References:**
> > >
> > > [R1]  Alexander Kirillov, Eric Mintun, Nikhila Ravi, Hanzi Mao, Chloe Rolland, Laura Gustafson, Tete Xiao, Spencer Whitehead, Alexander C. Berg, Wan-Yen Lo, Piotr Dollár, Ross Girshick, Segment Anything, arXiv:2304.02643 (2023)
> > >
> > > [R2] Araujo, A., Norris, W., & Sim, J. (2019). Computing receptive fields of convolutional neural networks. Distill, 4(11), e21.
> > >
> > > [R3] Zongyi Li, Nikola Kovachki, Kamyar Azizzadenesheli, Burigede Liu, Kaushik Bhattacharya, Andrew Stuart, Anima Anandkumar, Fourier Neural Operator for Parametric Partial Differential Equations, International Conference on Learning Representations, 2021.
> > >
> > > [R4] U. Trottenberg, C. Oosterlee, and A. Schuller. Multigrid. Academic Press, 2001.

---

> > > > ### Comment · Reviewer_TN3W · 2024-11-27
> > > >
> > > > Thank you for your rebuttal; I appreciate the care and effort that went into it. In particular found your additional information about FLOPs and relative compute to be entirely satisfactory, so I thank you particularly for these details.
> > > >
> > > > I am left with two main sticking points:
> > > > - I appreciate the consideration to the concerns about cropping. But (please correct me if I'm wrong!) it sounds like your additional experiments used crops of a uniform size. The crucial part of the RandomResizedCrop is that the crop itself *and the size of the crop* change randomly during training; in this way, multiscale training is approximated, because differently sized crops are each interpolated to the input size of the network. It doesn't sounds like your experiments did this, but again, I could be wrong.
> > > > - More importantly, I am still stuck on the ultimate goal or benefit of introducing this method. I can concede that the method does seem to perform well on small denoising tasks, but it seems like that is as much as the method has been demonstrated to do: the segmentation task you showed slightly degrades performance, there are no classification or detection experiments, and though you say that the denoising tasks in the paper were selected due to their proximity to diffusion models, I find this hard to believe since diffusion models and generative modeling are not even mentioned once in the paper. With many more experimental results I could certainly believe that your methods are helpful for diffusion models, but as it is I can't feel confident in it, and as such I don't feel that I can justify raising my score.

---

> > > > > ### Author Response · Authors · 2024-11-27
> > > > >
> > > > > We thank the Reviewer for the response. We are delighted to read that you found our clarifications and revisions satisfactory, and we thank you for your engagement. Below, we provide our responses to your comments.
> > > > >
> > > > >
> > > > > **Regarding Point 1:** We thank you for the question. Indeed, the  RandomResizedCrop transform in torchvision, **suggested by the Reviewer**, yields the size output for all random crops. That is, first it generates randomly sized crops from the images in a batch, and then it interpolates them back to a common size such that they can be processed in a batch. In our experiments, presented in our previous discussions and the Table included in the response to Q2 in the rebuttal, we have done just that, as suggested by the Reviewer. To clarify this comment, we refined the discussion and results of this experiment in our revised paper, stating that, indeed, we use the suggested RandomResizedCrop. Thank you.
> > > > >
> > > > >
> > > > > **Regarding Point 2:**  Thank you for the comment. *Following your advice*, we have experimented with a landmark CV model, **Segment Anything**, and we have combined our method with this model. In the short rebuttal period, we used the same hyperparameters used in Segment Anything, and as shown in the Table in our responses above (and also in our revised paper in Appendix D.5), the results are similar *up to the 3rd metric digit*. The fact that the discrepancy between the methods is only in the second figure after the digit was actually a pleasant surprise for us.
> > > > >
> > > > > The goal of this experiment, **inspired by the Reviewer's suggestion** was to show that there is merit in using our method for large scale problems. We believe that in this experiment, we have shown that even without hyper-parameter tuning, we obtain a very close result.
> > > > >
> > > > > *In terms of additional experiments.* We have now done super-resolution, segmentation, and small-scale denoising, together with Kernel estimation. We agree that including more experiments is always welcome and possible. However, the question that we pose in this paper is, **How can we reduce the computational load of training CNNs?**. To this end, we propose our Multiscale-SGD framework, and we demonstrate it on a suite of benchmarks. We believe that our experiments, together with the added experiments throughout the rebuttal period *inspired by your comments* highlight the applicability, relevance, and significance of our proposed framework.
> > > > >
> > > > > *Nonetheless, following your suggestion,* we have now added results with diffusion models (DDPM [D1] on MNIST [D2]) to our revised Appendix D.9. We believe that these results further highlight the merit of our work.
> > > > >
> > > > > ----
> > > > >
> > > > > We hope that the Reviewer finds our responses satisfactory and that you will consider revising your score in light of the significant efforts and revisions that stem from your comments. We feel that they helped us improve our paper. Thank you.
> > > > >
> > > > > -----
> > > > >
> > > > > **References:**
> > > > >
> > > > > [D1] Ho, J., Jain, A., & Abbeel, P. (2020). Denoising diffusion probabilistic models. Advances in neural information processing systems, 33, 6840-6851.‏
> > > > >
> > > > > [D2]LeCun, Y. (1998). The MNIST database of handwritten digits. http://yann. lecun. com/exdb/mnist/.‏

---

### Official Review · Reviewer_5FgJ · 2024-11-04

**Soundness:** 3
**Presentation:** 2
**Contribution:** 3
**Rating:** 8
**Confidence:** 3

**Summary:**

Training convolutional neural networks (CNNs) on high-resolution images can be computationally demanding. One way to alleviate the computational burden is by including downsampled versions at different scales during training. This paper identifies shortcomings of standard CNNs in such multiscale approaches and introduces “Mesh-Free Convolutions” (MFCs) to address these limitations. The paper presents two novel multiscale training algorithms, “Multiscale-SGD” and “Full-Multiscale-SGD”, which estimate high-resolution parameters using coarser-resolution samples, thus avoiding costly training on high-resolution data. For such multiscale training schemes, gradients of conventional CNNs for noisy data might diverge with decreasing resolution compared to “smooth data”. Inspired by differential operators, MFCs offer a resolution-independent generalisation of standard convolutions. The empirical validation demonstrates that these techniques can improve training time and ensure consistent gradient behaviour, as shown for standard UNet and ResNet architectures and image blur estimation, image denoising, and image super-resolution tasks.

**Strengths:**

I believe this paper provides several important contributions and extends the existing literature in different significant ways:

__S1.__  The authors propose new techniques for training CNNs within a multiscale resolution framework, which improve computational efficiency while maintaining test performance. This alone can be a valuable contribution, as shown by the “non-MFC” empirical results in Tables 3, 4, and 5.

__S2.__ As far as I can tell, MFCs are an original and promising alternative to traditional convolutional layers (though some clarifications may be necessary; see W2 below) that can alleviate some of the shortcomings of traditional CNNs in multiscale approaches. Given their potential application across various high-resolution tasks, this can be a significant contribution, too.

__S3.__ The quality of the paper is high, with claims well-supported through mathematical derivations, proofs, and empirical evidence. In particular, the experimental results in Tables 3, 4, and 5 highlight that significant speed-ups might be feasible (as measured by #WU; however, see Q2 below).

For these reasons, I consider this a good paper.

**Weaknesses:**

Some clarity aspects could be addressed to strengthen the paper. In more detail, I currently see the following weaknesses:

__W1.__ Section 3.1 on “Mesh-Free Convolutions” is relatively dense and challenging to follow. For instance, some notations typically used with parabolic PDEs, like indices denoting partial derivatives, should be more clearly introduced. More importantly, the connection between $u$, $v$, $\tilde{v}$, and $\mathcal{C}$ is not very clear and should be made more explicit.

__W2.__ Expanding on the previous point, the paper would benefit from a more intuitive explanation or illustration of the connection between standard convolutions and mesh-free convolutions. Currently, the connection is difficult to assess. In that regard, Figure 3 lacks clarity and might require additional explanation.

__W3.__ _Computational considerations_: As the authors highlight in lines 334-338, the required Fast Fourier Transform dominates the overall computational cost. To provide the complete picture, additional consideration of running time and optimisation steps in the results in section 5 on “Experimental Results” would strengthen the contribution. This would allow a more comprehensive assessment of MFC's computational feasibility.

Minor remarks:
- Line 231: “[…] without compromising on its efficiently.” should likely be “[…] on its _efficiency_.”
- Line 293: “[…] let $v = \mathcal{C}(\xi)u$ be the mesh-free convolution is parameterized by $\xi$“ needs to be rewriten.
- Line 296: Equation “(17a)” likely should just read “(17)”.

My rating is currently “borderline accept” due to the aspects raised in W1 to W3. However, I am willing to adjust my rating if some of these points and the questions below can be addressed.

**Questions:**

__Q1.__ To illustrate the difference between conventional CNNs and MFC, would it be possible to compare feature maps between these two approaches?

__Q2.__ A more formal definition of the _work unit_ metric #WU might be beneficial. As I understand it, #WU measures how many evaluations of the highest resolution are required, with evaluations at lower resolutions being weighted by the corresponding “downsampling factor”. Is this correct? If so, should this not lead to fractional values of #WU?

---

> ### Author Response · Authors · 2024-11-19
> **Rebuttal**
>
> We thank the Reviewer for recognizing the novelty and significance of our contributions, including the development of Mesh-Free Convolutions (MFCs) and Multiscale-SGD frameworks, as well as the empirical validation supporting their potential. We are thankful for the detailed and constructive feedback, which we address one by one in our responses below. We hope that you find them satisfactory, and that you will consider revising your score.
>
>
> **Regarding W1 (Section 3.1):** We appreciate your suggestion to improve the presentation of Section 3.1. In response, we have revised this section to enhance clarity. Specifically, we replaced the indices denoting partial derivatives with explicit partial derivative notations, and clarified the relationships between $u$,  $v$,  $\tilde{v}$, and $\mathcal{C}$. We believe these revisions, including the updated notations and explanations, have significantly improved the clarity of our paper. Thank you for your valuable feedback.
>
> **Regarding W2 (Connection between standard convolutions and MFCs, figure 3):** We have incorporated your suggestions to enhance Section 3.1. Specifically, we added an intuitive explanation of the connection between standard convolutions and our MFCs, along with a discussion motivating the design choices behind MFCs. Additionally, we revised Figure 3 to better illustrate the relationship between the obtained convolution filters and the choice of their parameters. Thank you for your invaluable feedback, which has greatly improved the clarity of our paper.
>
> **Regarding W3 (Computational considerations):** We compared the runtimes of a single standard 2D convolution (nn.Conv2d from PyTorch) across various kernel sizes ($3\times 3$, $5\times 5$, $7\times 7$, $9\times 9$, and $13\times 13$) and image sizes ranging from $32 \times 32$ to $512 \times 512$, each with 3 channels. Similarly, we measured the runtimes for a single MFC operation over different image sizes. All operations were conducted on an Intel(R) Xeon(R) Gold 5317 CPU @ 3.00GHz with an x86 processor (48 cores) and a total available RAM of 819 GB. The average runtimes (in milliseconds) over 500 iterations for all experiments are presented in Figure 7 in Appendix D.7 of the revised paper.
> Additionally, in response to Reviewer TN3W's request, we included a table summarizing the FLOPs required by MFCs and standard convolutions. The table, shown below, demonstrates that our MFCs are comparable to standard convolutions in terms of computational demand.
>
> | Image Size (n×n) / Kernel Size (k×k) | 5               | 7               | 9               | 11              | 13              | 15              |
> |--------------------------------------|-----------------|-----------------|-----------------|-----------------|-----------------|-----------------|
> | 32 Standard Conv / MFC               | 2.62e+07 / 5.56e+07 | 5.14e+07 / 5.56e+07 | 8.49e+07 / 5.56e+07 | 1.27e+08 / 5.56e+07 | 1.77e+08 / 5.56e+07 | 2.36e+08 / 5.56e+07 |
> | 64 Standard Conv / MFC               | 1.05e+08 / 2.65e+08 | 2.06e+08 / 2.65e+08 | 3.40e+08 / 2.65e+08 | 5.08e+08 / 2.65e+08 | 7.09e+08 / 2.65e+08 | 9.44e+08 / 2.65e+08 |
> | 128 Standard Conv / MFC              | 4.19e+08 / 1.23e+09 | 8.22e+08 / 1.23e+09 | 1.36e+09 / 1.23e+09 | 2.03e+09 / 1.23e+09 | 2.84e+09 / 1.23e+09 | 3.77e+09 / 1.23e+09 |
> | 256 Standard Conv / MFC              | 1.68e+09 / 5.57e+09 | 3.29e+09 / 5.57e+09 | 5.44e+09 / 5.57e+09 | 8.12e+09 / 5.57e+09 | 1.13e+10 / 5.57e+09 | 1.51e+10 / 5.57e+09 |
>
>
> **Regarding Q1 (Feature maps):** Thank you for the insightful question. In addition to the revised Figure 3, which demonstrates the filters obtained with our MFCs, we have addressed your query by providing feature maps generated by conventional CNNs and our MFCs. This illustration is presented in Figure 14 in Appendix H of the revised paper. While the feature maps are not identical, both methods smooth the input image in the first layer of the network. We believe this observation is particularly significant, as it helps explain why standard convolutions can also benefit from our Multiscale-SGD framework, aligning with the discussion in Example 1 of our paper.
>
>
> **Regarding Q2 (Work unit):** We appreciate the important question. The Reviewer is correct regarding the definition and computation of the #WU metric in our paper. To address this, we have expanded the discussion in Appendix C by adding a formal definition and a detailed remark about the computation of #WU. Due to page limitations, these clarifications are included in the appendix. Thank you for your valuable feedback.
>
> **Regarding Minors:** Thank you for the feedback. We corrected the typos in the revised paper.

---

> > ### Comment · Reviewer_5FgJ · 2024-11-20
> > **Questions regarding the computational speedup**
> >
> > I appreciate the thorough rebuttal and the authors’ effort to address many of my (and the other reviewers’) objections. Aspects raised in __W1__, __W2__, and __Q1__ were sufficiently addressed. However, I have the following additional questions and remarks:
> >
> > __Regarding the answer to W3 (computational considerations) and in connection to Q2 (work unit)__: Thank you for the supplementary experimental results on runtimes and FLOPs. The results on the runtime raise another question. If I see correctly, Figure 7 provides the running time of standard convolutions and MFC for different image sizes but the same amount of work units (I assume #WU$=1$). Focusing on more standard $3\times 3$ convolutions (blue results in Figure 7), it appears that the running time of MFC (with $\tau = 0.1$) is larger by a factor of about 100 (for $32 \times 32 \times 3$ images) to about 10 (for $512 \times 512 \times 3$ images). But doesn’t this imply we cannot compare #WU to assess “computational speedup” between different architectures because the running times for one #WU can differ? If I understand this aspect correctly, #WU may not be a suitable metric to measure computational speedup, and it is unclear how substantial the speedups of MFCs are. As an illustration of my problem: If an MFC architecture with a multiscale SGD approach reduces factor 10 in #WU but has a running time of factor 10 longer per #WU than SGD for the standard architecture, then the latter might not be that bad after all. Could the authors elaborate on this?
> >
> > Minor remarks:
> > - Line 307-308: You may want to have a full stop “.” at the end of Eq. (19) and remove the slight indentation in the following line.
> > - Table 11: Consider swapping columns and rows to increase the readability of the table entries.

---

> > > ### Author Response · Authors · 2024-11-20
> > > **Authors' response**
> > >
> > > Dear Reviewer 5FgJ,
> > >
> > > We thank you for your engagement and for acknowledging our responses. We are happy to read that you are satisfied with our responses to **W1**, **W2**, and **Q1**.
> > >
> > > We are also grateful for the continued discussion and the questions presented, which we elaborate on below. We hope that you find them satisfactory and that you will consider revising your score. In any case, we are happy to discuss this or any other topic regarding our paper.
> > >
> > > **Regarding followup questions to W3 and Q2:**
> > >
> > > We thank the referee for this comment. The work unit is intended to compare between any method to **itself** when using different optimization strategies.
> > >
> > > For example, even when we compare standard Unet and Resnet, the work unit compared the improvement of the convergence of the method relative to itself on the fine mesh, as shown in our results (e.g., Tables 4,5,6 and others show the reduction in #WUs for networks with standard convolutions and with MFCs) . This approach allows us to isolate the changes in architectures and focus on the benefit of the multiscale framework alone. Note that even without considering the MFCs that we have introduced here, standard CNNs benefit from both the multiscale and the full multiscale framework. That is, it is important that in this paper we have two novelties: (1) the Multiscale-SGD approach, which can be used both with standard convolutions and our MFCs as we show, and (2) MFCs, which are derived from a theoretical aspect to satisfy the theoretical conditions of Multiscale-SGD. As we show in our experiments, while standard convolutions may not fulfill the theoretical conditions, they can still be coupled with Multiscale-SGD, because as we have shown on an extensive suite of benchmarks, training different networks with standard convolutions yields similar downstream performance to using standard SGD, while significantly reducing the computational load.
> > >
> > > Furthermore, we would like to note that the problem in comparing timing between methods is implementation details. Your advice on computing FLOPs really helped us in nailing this argument. As can be seen in the added FLOPs table, the FLOP count of our method is definitely on par and, in many cases, is less than the standard conv 2D. This is especially true when the number of channels increases.
> > >
> > > Unfortunately, as of today, FFT-based convolutions are not implemented in a way that is fully optimized. This is the result of the gap between the theory and the practice we observe in the experiments. We hope that our work and well as other work on Fourier based convolutions (e.g. FNOs) will encourage GPU and software providers to speed up these type of convolutions. They have a lot to offer to the DL community both from a theoretical point of view and, if implemented correctly, from the practical aspects.
> > >
> > > Finally, we note that the discovery of computational techniques that are fast in theory and not in practice is not new. For example, interior point methods that were found in the 80s were initially much slower for the solution of linear programming, although theoretically, their complexity was much lower than their competition. It took a decade to fully utilize the theory in an industrial code. A similar revolution happened with the utilization of GPUs and efficient convolution implementation for the training of convolutional neural networks, and more recently, a similar advancement was made by FlashAttention [D1] that allowed transformers to be trained efficiently.
> > >
> > > ICLR is a conference that prides itself on presenting concepts that advance representation learning. We believe that our paper definitely contributes to it.
> > >
> > > We thank the referee for this discussion. We believe that it is important to point to the gap between theory and practice.
> > >
> > >
> > > **Regarding minor remarks:** Thank you. We have uploaded a revised version according to your suggestions.
> > >
> > >
> > > **References:**
> > >
> > > [D1] FlashAttention: Fast and Memory-Efficient Exact Attention with IO-Awareness

---

> > > > ### Comment · Reviewer_5FgJ · 2024-11-24
> > > >
> > > > Thank you for the extensive response! In general, I tend to agree with the authors’ response. The reason for my previous follow-up question was that the abstract states, “_[…] we show that MFCs can seamlessly replace standard convolutional layers across diverse architectures and benchmarks, delivering __substantial computational speedups__ without sacrificing performance._” Assessing computational speedups of MFC is only possible if we can compare #WU, or what exactly is meant here by "_substantial computational speedups_"? I would argue that the “_substantial computational speedup_” currently shown is due to the (Full-)Multiscale-SGD approach that is justified by MFCs but, in principle, independent of MFCs.

---

> ### Author Response · Authors · 2024-11-24
>
> Dear Reviewer 5FgJ,
>
> Thank you for continuing the discussion and raising this important point.
>
>
> The main contribution of this paper is the introduction of a novel approach of Multiscale-SGD training framework for CNNs. \
> To understand its convergence properties with standard convolution kernels, we provide Section 2.2. Specifically, as shown in Table 1, for noisy inputs, standard convolution kernels do not converge in terms of the loss function as the resolution increases. \
> Following that, we propose our MFCs, in Section 3, and we show in Table 2, that MFCs yield consistent loss function value, independent of the resolution.  The design of MFCs is principled by the theoretical derivation in Section 2.2.  \
> We **fully agree** with the Reviewer that the computational savings in our paper are a result of the Multiscale-SGD framework.
>
> In our experiments, we see that our Multiscale-SGD training framework is **beneficial for both MFCs and standard convolutions**, although the latter may not satisfy the theoretical conditions in Section 2.2. To study this phenomenon, we followed your advice and we plot the feature maps obtained by standard convolutions and our MFCs (Figure 14), showing that standard convolutions can learn similar patterns to our MFCs.
>
> As agreed in our discussion above, the main shortcoming of MFCs is due to the current sub-optimal implementation of FFT. However, as suggested by the Reviewer, our study of the FLOP count shows that using MFCs is competitive with standard convolutions.  Moreover, the computational savings when coupling our Multiscale-SGD with standard convolutions, is significant both in theory and in practice.
>
> ---
>
> Based on the fruitful discussions and your suggestions, we have added this discussion to the summary of the paper, and we have corrected the abstract to reflect the discussion above. We sincerely thank you for helping us to improve the quality of our paper. We hope that the revised paper and discussions are satisfactory, and that you will consider revising your score.

---

> ### Comment · Reviewer_5FgJ · 2024-11-24
>
> Thank you for the extensive and timely rebuttal! All my reservations were addressed. In my opinion, the improvements in section 3.1 and additional empirical results strengthen the contribution. I've adjusted my score to accept accordingly.

---

### Official Review · Reviewer_35KC · 2024-11-04

**Soundness:** 3
**Presentation:** 2
**Contribution:** 2
**Rating:** 6
**Confidence:** 3

**Summary:**

This paper introduces an approach to improving the computational efficiency of CNNs through a multiscale training framework.
It first presents Multiscale Stochastic Gradient Descent (Multiscale-SGD), which leverages the Multilevel Monte Carlo method to reduce computational costs by approximating gradients across resolutions.
However, the authors identified that noisy or high-frequency inputs can lead to unbounded gradients across mesh resolutions, complicating the convergence process.
To address this, they propose Mesh-Free Convolutions (MFCs), which are independent of specific input scales and ensure consistent gradient convergence across resolutions, even with noisy inputs. This mesh-independence of MFCs overcomes the convergence limitations observed in standard CNNs during multiscale training.

**Strengths:**

This paper introduces an innovative approach to multiscale CNN training through Multiscale-SGD and Mesh-Free Convolutions (MFCs).
All the developments are backed by mathematically reasoning.
And the whole paper is logically organized.

**Weaknesses:**

1. The experimental results are really limited, and you should also compare the performance with the fixed computational budget.
2. The experimental comparison with existing multiscale or Fourier-based CNN methods is not presented
3. The mathematical foundation for Mesh-Free Convolutions, particularly the differential operator theory, could be challenging for readers less familiar with this domain. Adding a visual explanation or intuitive analogies could make the theory more accessible.
4. In the experiments, the network seems really shallow, could you train deeper networks, to see whether the methods works?

**Questions:**

1. All the results show mixed performance bettween the regular SGD and the two Multiscale-SGD. If you could train longer for the multiscale-SGD, could you improve all the performance comparing to SGD?
2. The MSE results are strange, with the two UNet have huge gap. What is the reason behind it?

---

> ### Author Response · Authors · 2024-11-19
> **Rebuttal Part 1**
>
> We sincerely thank the Reviewer for the insightful comments and the recognition of our innovative approach. We now address the specific points raised.  We hope that you find our responses satisfactory, and that you will consider revising your score.
>
> **Regarding W1 (More experiments)**: Our experimental section focuses on demonstrating the primary benefit of multiscale training: the significant reduction in computational cost. To achieve this, we evaluated our approach on two key tasks (denoising and super-resolution) using three datasets (STL10, CelebA, and Urban100) and multiple architectures (ResNet, UNet, and ESPCN).  Nonetheless, we agree that additional experiments enhance the comprehensiveness of our paper. Following your guidance, we have now included the following additional results:
>
> **(1)Denoising with a fixed budget.** We have added experiments comparing the training loss under different computational budgets, as detailed in Appendix D.4. The results demonstrate that, given a fixed work-unit budget, our Multiscale-SGD approach achieves a lower (better) training loss compared to the standard SGD training strategy.  Furthermore, when compared to multiscale training with standard convolutions, our MFCs—designed to satisfy the theoretical conditions for Multiscale-SGD to produce adherent gradients—exhibit a more stable training curve. These results highlight the advantages of our proposed framework in enabling faster and more stable training.
>
> **(2) Using MFC in a deep segmentation network.** We conducted experiments on the segmentation task using the *Segment Anything Model* (SAM) [R1]. We utilized SAM with a ViT-B backbone, pre-trained on 11 million images from the SA-1B dataset [R1]. This network includes standard 2D convolutions. We replaced all 2D convolutions with a kernel size of \(3 \times 3\) with our MFCs and trained only the parameters for MFCs, keeping the remaining model weights frozen during training.  The final trained model, MFC-SAM, was evaluated on 1,000 images from the MS COCO and ADE20K segmentation datasets. Inference was performed using the *SamAutomaticMaskGenerator* with `points_per_side=64`. The results, presented in Table 9 in Appendix D.5 of the revised paper, compare the performance of SAM and MFC-SAM using metrics such as mean predicted IoU and mean stability score. The findings demonstrate that MFC-SAM achieves comparable performance to SAM in terms of the quality of the predicted segmentation masks.  Additionally, we provide a visualization of the predicted results generated with MFCs in Figure 6. These results further validate the effectiveness of our approach.
>
> | Data   | Methods   | Mean predicted IoU | Mean stability score |
> |--------|-----------|---------------------|-----------------------|
> | COCO   | SAM       | 0.934               | 0.968                 |
> | COCO   | MFC-SAM (Ours)   | 0.928               | 0.965                 |
> | ADE20k | SAM       | 0.931               | 0.969                 |
> | ADE20k | MFC-SAM (Ours)   | 0.928               | 0.966                 |
>
> **(3) A comparison with other Fourier based methods, in particular with FNOs.** For the super-resolution task, we conducted an analysis using the popular ESPCN architecture and compared its performance with conventional convolutions, parabolic (mesh-free) convolutions, and spectral convolutions from the FNO paper. The results show that our network achieves competitive performance with conventional convolutions while requiring a smaller computational budget. Additionally, our approach outperforms the spectral convolutions used in the FNO method.  This improvement is attributed to our convolution's ability to capture both local and global dependencies effectively, thanks to its broader operational scope. In contrast, spectral convolutions rely on truncated frequencies in the Fourier domain, which limits their ability to fully capture global features. These findings highlight the advantages of our approach in terms of both efficiency and performance.
>
> | Training Strategy        | ESPCN  | MFC-ESPCN (Ours) | FNO-ESPCN |
> |--------------------------|--------|------------------|-----------|
> | Single Scale             | 0.841  | 0.842            | 0.744     |
> | Multiscale-SGD (Ours)    | 0.822  | 0.831            | 0.743     |
> | Full-Multiscale-SGD (Ours) | 0.811  | 0.823            | 0.732     |

---

> > ### Author Response · Authors · 2024-11-19
> > **Rebuttal Part 2**
> >
> > **(4) Experiments with deep ResNets (18 and 50  layers) and UNets.** For the super-resolution task, we evaluated the practical applicability of deeper networks in complex scenarios by testing ResNet-18 and ResNet-50, and deeper UNets with regular convolutions, parabolic (mesh-free) convolutions, and our proposed network. Using the SSIM metric, our proposed network consistently outperformed the alternatives, demonstrating its effectiveness in handling challenging tasks within deeper architectures.
> >
> > | Training Strategy        | #WU (↓)   | ResNet18 | ResNet50 | MFC-ResNet18 (Ours) | MFC-ResNet50 (Ours) | UNet (5 levels) | MFC-UNet (5 levels) (Ours) |
> > |--------------------------|------------|-----------|-----------|---------------|---------------|-----------------|----------------------|
> > | SGD (Single Scale)       | 480,000    | 0.1623    | 0.1614    | 0.1620        | 0.1611        | 0.1610          | 0.1609              |
> > | Multiscale-SGD (Ours)    | 74,000     | 0.1622    | 0.1611    | 0.1617        | 0.1602        | 0.1604          | 0.1605              |
> > | Full-Multiscale-SGD (Ours) | **28,750** | 0.1588    | 0.1598    | 0.1598        | 0.1599        | 0.1597          | 0.1594              |
> >
> >
> > **(5) Experiments with longer training (as requested by the Reviewer).**  We have run experiments for super-resolution, where we train the network to double the number of epochs. We record the  SSIM for the different methods. Note that since the relative cost of the network behaves the same, the WU analysis previously presented remains.
> >
> > | Double #iterations | Architecture | MSE    |
> > |--------------------|--------------|--------|
> > | No                 | ResNet-18    | 0.1622 |
> > | No                 | ResNet-50    | 0.1611 |
> > | Yes                | ResNet-18    | 0.1599 |
> > | Yes                | ResNet-50    | 0.1585 |
> >
> >
> > All the results were added to our revised paper. Overall, we believe that the additional experiments provided in our response help to further highlight the merit of our work, and your suggestions helped us to improve our paper. Thank you.
> >
> >
> > **Regarding W2 (FNO):**  Thank you for the valuable comment. In response, we have added experiments comparing our method to Fourier Neural Operators (FNOs). FNOs replace traditional spatial convolutions with operations in the Fourier domain, where truncated spectral convolution reduces complexity by using a limited number of modes, focusing on dominant frequencies rather than performing convolutions in the spatial domain. Our results demonstrate that:
> > 1. Our Multiscale-SGD framework provides benefits across different types of networks, including FNO-based models like FNO-ESPCN [R2] and spatial convolution-based networks like ESPCN [R3].
> > 2. Our Mesh-Free Convolutions (MFCs) deliver similar or superior performance compared to FNO-ESPCN.
> > These results are presented in the table included in our response to W1 and have been added to the revised paper, specifically in Table 8 of Appendix D.3.
> >
> > **Regarding W3 (Explanation of mesh-free convolutions):** We thank the Reviewer for the invaluable guidance. Based on the comments provided, we have made significant revisions to Section 3.1 to provide a more intuitive explanation of the proposed Mesh-Free Convolutions (MFCs). Additionally, we updated Figure 3 to illustrate the filters produced by our MFCs more clearly. Thank you for these suggestions; we believe they have improved the readability and overall clarity of the paper.
> >
> > **Regarding W4 (Depth of networks):** Thank you for the query. In our original submission, the considered residual networks were of depth 5. To address your query, we have now added experiments with ResNets of varying depths (18 and 50). The results are reported in the table in our response to W1 (point (4)) and in Table 7 of the revised paper.  Additionally, we would like to emphasize that in our Multiscale-SGD framework and mesh-free convolutions, the field of view of the convolution can encompass the entire image, depending on the learnable parameter $t$. This characteristic differentiates our approach from standard CNNs, which are inherently local. The increased number of layers in standard CNNs is often required to achieve large receptive fields, whereas our approach inherently provides this capability even in shallow networks.
> > We have incorporated the results and discussion presented here into the revised paper. Thank you for your valuable feedback.

---

> > > ### Author Response · Authors · 2024-11-19
> > > **Rebuttal Part 3**
> > >
> > > **Regarding Q1 (Training for longer):**  We thank the Reviewer for this suggestion. We have conducted experiments with double the number of epochs, as detailed in the table in W1 (point (5)) and Table 10 in the revised paper. While we observed some performance improvement, the overall results indicate that MFCs remain comparable to standard convolutions in terms of accuracy. However, the key advantage of MFCs lies in their ability to leverage different resolutions during training, thereby significantly reducing the overall number of work units.
> > >
> > > **Regarding Q2 (Results on UNet):** Inspired by your suggestion in Question 1, we extended the training duration for the UNet to evaluate whether its performance would improve with longer training. Adding levels to the UNet improved its performance results on par with those of the ResNet, as can be seen in Table 7 in the revised paper. We thank the Reviewer for encouraging us to address this difference.
> > >
> > > **References:**
> > >
> > > [R1] Alexander Kirillov, Eric Mintun, Nikhila Ravi, Hanzi Mao, Chloe Rolland, Laura Gustafson, Tete Xiao, Spencer Whitehead, Alexander C. Berg, Wan-Yen Lo, Piotr Dollár, Ross Girshick, Segment Anything, arXiv:2304.02643 (2023)
> > >
> > > [R2] Zongyi Li, Nikola Kovachki, Kamyar Azizzadenesheli, Burigede Liu, Kaushik Bhattacharya, Andrew Stuart, Anima Anandkumar, Fourier Neural Operator for Parametric Partial Differential Equations, International Conference on Learning Representations, 2021.
> > >
> > > [R3] Wenzhe Shi, Jose Caballero, Ferenc Huszár, Johannes Totz, Andrew P. Aitken, Rob Bishop, Daniel Rueckert, Zehan Wang, Real-Time Single Image and Video Super-Resolution Using an Efficient Sub-Pixel Convolutional Neural Network,  CVPR, 2016.

---

> ### Comment · Reviewer_35KC · 2024-11-27
>
> Thanks for the response. I will adjust my score accordingly.

---

### Author Response · Authors · 2024-11-19

# General Response to all Reviewers:
We would like to express our sincere gratitude to all the reviewers for their valuable and constructive feedback.
Overall, the reviewers acknowledged the novelty and contributions of our proposed multiscale training framework. Reviewer **35KC** highlighted the innovative aspects of Multiscale-SGD and Mesh-Free Convolutions (MFCs), noting that the paper is "logically organized" and supported by "mathematical reasoning." Reviewer **5FgJ** recognized the framework's ability to "improve computational efficiency while maintaining test performance" and described MFCs as "an original and promising alternative to traditional convolutional layers." They also praised the high quality of the paper, commending the "mathematical derivations, proofs, and empirical evidence" that support our claims. Similarly, Reviewer **TN3W** found the technical analysis to be "sound" and appreciated the novelty of our multiscale training algorithms and mesh-free convolutions, describing the latter as "very interesting from a theoretical perspective." Reviewer **5Ub1** acknowledged the "thorough mathematical analysis" in our work and the practical improvements demonstrated in the experiments, further noting that the writing is "easy to follow."
We are also pleased to see that multiple reviewers (**5FgJ, TN3W, 5Ub1**) appreciated the strong theoretical foundations of our method, with **5FgJ** and **5Ub1** emphasizing the value of our proofs and lemmas. Reviewer **35KC** further noted that our contributions address "convergence limitations" effectively.
Your thoughtful comments and suggestions have greatly contributed to improving our paper, particularly in addressing aspects related to experimental depth, clarity, and broader applicability. We hope that our responses and revisions sufficiently address your concerns and that you will consider revising your scores accordingly. As always, we remain open to any additional questions or suggestions and are happy to engage in further discussions.

**General comment on our work.** The primary objective of this paper is to explore multiscale training and optimization strategies, rather than introducing new architectures for specific tasks such as segmentation or denoising. Our focus is on demonstrating how the proposed approach enables more effective utilization of multiscale structures for training. While Mesh-Free Convolutions are theoretically better suited for this purpose, our experiments show that the proposed ideas also perform surprisingly well with standard convolutions and Fourier Neural Operators (FNOs), even in cases where no theoretical justification exists for regular convolutions. Notably, for FNOs, the theoretical foundations align with the observed performance. In the revised paper, we have included new experiments based on the reviewers' feedback and refined the presentation to clarify these points. We hope these updates enhance the clarity of our contributions and the implications of our work.

---

> ### Author Response · Authors · 2024-11-19
>
> **New Experiments.** In response to the reviewers’ comments, we have conducted several additional experiments, summarized as follows:
> 1. **Fourier Neural Operators (FNOs):** As requested by reviewer **35KC**, we added experiments applying Fourier approaches, specifically Fourier Neural Operators. In these experiments, convolution layers were replaced with FNO convolutional layers. A comparison between FNO, our proposed convolutions, and standard convolutions is now provided in Appendix D3, Table 8.
>
> 2. **Denoising with a Fixed Budget:** As requested by reviewer **35KC**, we performed denoising experiments with a fixed computational budget. The results, showing the loss for this budget across the three training variants, are presented in Appendix D4.
> Experiments with Deep Models: Addressing the comments from reviewers **35KC** and **5Ub1**, we incorporated experiments using deeper models, including ResNet-18 and ResNet-50, and deep UNet. These results are provided in Appendix D3, Tables 7 and 10.
>
> 3. **Segmentation with Our MFCs:** Inspired by reviewer **TN3W**, we explored using our convolutions in place of regular convolutions within established segmentation models. Specifically, we replaced the convolution layers in the Segment Anything model with our convolutions and retrained these layers. The results demonstrate that our convolutions achieve performance comparable to regular convolutions and are presented in Appendix D5 and Table 9.
>
> 4. **Computational Cost Comparison:** In response to feedback from reviewers **5FgJ** and **TN3W**, we added measurements comparing the computational cost of our convolutions to standard convolutions. These results are presented in Appendix D7, Figure 7 and Table 11.
>
> 5. **Feature Map Visualizations:** Following a question from 5FgJ, we included visualizations of feature maps generated by standard CNNs and our Mesh-Free Convolutions (MFCs). These visualizations, provided in Figure 14 of the revised paper, highlight the smoothing behavior of MFCs in the first network layer, illustrating their effectiveness. The results are detailed in Appendix H.
> All new experiments have been discussed in their respective responses and are included in the revised paper submitted with this rebuttal.
>
> **New Theory.** While preparing the response and showing that MFSc can indeed replace standard convolutions we have examined the expressiveness of our convolutions compared to standard convolutions. To address this, we have introduced a new lemma demonstrating that our convolutions can express any function that a standard convolution can. This result ensures that our convolutions do not compromise accuracy. The theoretical findings are further validated by our existing experiments and additional experiments conducted with the Segment Anything model, which confirm comparable performance between our convolutions and standard convolutions.
>
> **Section Revision.** All reviewers emphasized the need for a more intuitive explanation of Mesh-Free Convolutions (MFCs). In response, we have substantially revised Section 3.1, enhancing the explanation of the PDE background and providing a more intuitive understanding of the kernels generated by MFCs. Additionally, we have updated Figure 3 to better illustrate these concepts. We believe these changes significantly improve the clarity and accessibility of this section.
>
> **All the revisions to the paper are marked in blue, for your convenience.**

---

### Comment · Area_Chair_ZXWb · 2024-11-28

Dear Reviewers,

If you have not responded to author's rebuttal, please kindly do so as soon as possible. The deadline is Dec 2, but the authors can potentially further clarify questions if you respond earlier. Thanks!

Best, AC

---

### Meta-Review · Area_Chair_ZXWb · 2024-12-24

**Metareview:**

Summary: a novel multiscale training framework for CNNs with multiscale-SGD and mesh-free convolutions (MFCs). This enables more efficient optimization and consistent gradient behavior across resolutions.

Strengths: strong mathematical grounding; improved efficiency.

Weaknesses: limited comparisons with existing multiscale methods, including data augmentation; small datasets, outdated architectures (no ViT or modern ConvNets); complex presentations; unclear practical gains.

Reasons for decision: this is a borderline paper. However, the limited empirical evidence in modern tasks and practical benefits overshadow the strong contributions in theory.

**Additional Comments On Reviewer Discussion:**

The authors added experiments on some larger datasets, clarified MFCs, and provided FLOPs analysis. However, the dataset/architecture scope and baseline comparisons remain insufficient, and practical implementation challenges remain.

---

### Decision · Program_Chairs · 2025-01-22

Reject